# High-throughput neutralization measurements correlate strongly with evolutionary success of human influenza strains

Caroline Kikawa[1,2,3†], Andrea N Loes[1,4†], John Huddleston[5], Marlin D Figgins[1,5], Philippa Steinberg[5], Tachianna Griffiths[6], Elizabeth M Drapeau[6], Heidi Peck[7], Ian Barr[7], Janet A Englund[8,9], Scott E Hensley[6], Trevor Bedford[4,5], Jesse D Bloom[1,2,4]*

[1]Division of Basic Sciences and Computational Biology Program, Fred Hutch Cancer Center, Seattle, United States; [2]Department of Genome Sciences, University of Washington, Seattle, United States; [3]Medical Scientist Training Program, University of Washington, Seattle, United States; [4]Howard Hughes Medical Institute, Seattle, United States; [5]Vaccine and Infectious Disease Division, Fred Hutch Cancer Center, Seattle, United States; [6]Department of Microbiology, Perelman School of Medicine, University of Pennsylvania, Philadelphia, United States; [7]WHO Collaborating Centre for Reference and Research on Influenza, The Peter Doherty Institute for Infection and Immunity, Melbourne, Australia; [8]Seattle Children's Research Institute, Seattle, United States; [9]Department of Pediatrics, University of Washington, Seattle, United States

*For correspondence:
jbloom@fredhutch.org

†These authors contributed equally to this work

## eLife Assessment

This **fundamental** study advances our understanding of population-level immune responses to influenza in both children and adults. The strength of the evidence supporting the conclusions is **compelling**, with high-throughput profiling assays and mathematical modeling. The work will be of interest to immunologists, virologists, vaccine developers, and those working on mathematical modeling of infectious diseases.

**Abstract** Human influenza viruses rapidly acquire mutations in their hemagglutinin (HA) protein that erode neutralization by antibodies from prior exposures. Here, we use a sequencing-based assay to measure neutralization titers for 78 recent H3N2 HA strains against a large set of children and adult sera, measuring ~10,000 total titers. There is substantial person-to-person heterogeneity in the titers against different viral strains, both within and across age cohorts. The growth rates of H3N2 strains in the human population in 2023 are highly correlated with the fraction of sera with low titers against each strain. Notably, strain growth rates are less correlated with neutralization titers against pools of human sera, demonstrating the importance of population heterogeneity in shaping viral evolution. Overall, these results suggest that high-throughput neutralization measurements of human sera against many different viral strains can help explain the evolution of human influenza.

**eLife digest** Pathogens are in a constant evolutionary battle to outwit the immune system. For example, influenza viruses rapidly mutate their hemagglutinin protein to evade antibody proteins generated by past infections or vaccinations. This is one reason why people can be repeatedly infected by influenza viruses throughout their lives.

Understanding this process is crucial for selecting the right viral strains for the annual flu vaccine. However, interpreting the evolution of the flu virus is challenging because individuals vary greatly in their exposure histories, resulting in diverse antibody repertoires that target the virus in different ways. Kikawa et al. developed a new method to measure antibody levels against 78 recent flu strains in both children and adults. They discovered that individuals differed significantly in their ability to neutralize various strains. Furthermore, in 2023, viral strains with a higher proportion of people having lower antibody levels appeared to grow much more rapidly in the human population. This suggests that measuring antibody levels against current viral strains can help identify which ones should be included in next year's vaccine.

Overall, the study by Kikawa et al. shows that experimental approaches capturing the complexity of human immune responses can provide valuable insights into the evolution of flu viruses. Such insights can improve the selection of strains for the annual flu vaccine.

## Introduction

Infection or vaccination with influenza virus elicits a neutralizing antibody response targeting the viral hemagglutinin (HA) protein (*Couch and Kasel, 1983*; *Krammer, 2019*). These antibodies correlate with protection against infection and so provide substantial immunity to strains that they neutralize (*Couch and Kasel, 1983*; *Krammer, 2019*; *Couch, 1975*; *Hobson et al., 1972*; *Davies et al., 1982*; *Yu et al., 2008*; *Tsang et al., 2014*; *Fox et al., 1982*; *Coudeville et al., 2010*; *Ng et al., 2013*; *Ohmit et al., 2011*). However, the HA of human influenza evolves rapidly, acquiring mutations that erode neutralization by antibodies elicited by prior infections and vaccinations (*Bedford et al., 2014*; *Smith et al., 2004*; *Doud et al., 2017*). New HA variants with reduced neutralization are generally the most evolutionarily successful and repeatedly replace the current dominant variant(s) in a process known as antigenic drift (*Smith et al., 2004*; *Hay et al., 2001*; *Neher et al., 2016*; *Kim et al., 2024*). As a result, people are reinfected roughly every 5 years (*Couch and Kasel, 1983*; *Kucharski et al., 2015*; *Ranjeva et al., 2019*), and vaccines are updated annually to attempt to match the currently dominant influenza strains (*Gerdil, 2003*).

The human antibody response to influenza is shaped in part by past exposures to related strains, a phenomenon known as imprinting (*de St.Groth and Webster, 1966*; *Francis, 1960*; *Cobey and Hensley, 2017*). While an individual's exposure history is partially dependent on their birth year (*Ranjeva et al., 2019*; *Lessler et al., 2012*; *Miller et al., 2013*), HA's rapid evolution along with variation in which strains infect even individuals with similar birth years create heterogeneous exposure histories both between and within birth cohorts (*Krammer, 2019*; *Francis, 1960*; *Linderman et al., 2014*; *Fonville et al., 2014*; *Skowronski et al., 2017*). These heterogeneous exposure histories lead to substantial differences in how the neutralizing antibodies of different people target HA, and hence how their neutralizing antibody immunity is affected by viral mutations (*Linderman et al., 2014*; *Welsh et al., 2024*; *Lee et al., 2019*).

Determining how population heterogeneity in human neutralizing antibody specificities shapes influenza virus evolution has been challenging because conventional methods (e.g. hemagglutination inhibition *Hirst, 1943* and neutralization assays *Okuno et al., 1990*) are low throughput. It is therefore experimentally daunting to use these methods to measure neutralizing titers for large numbers of human sera against the full diversity of influenza virus strains that circulate in a single season. In part because of these limitations, a common approach is to use sera from singly infected ferrets to estimate antigenic distances between different viral strains (*Smith et al., 2004*; *Jorquera et al., 2019*). However, growing evidence indicates sera from singly infected ferrets are an imperfect proxy for human populations with complex and heterogeneous exposure histories (*Cobey and Hensley, 2017*; *Linderman et al., 2014*; *Fonville et al., 2014*; *Lee et al., 2019*; *Li et al., 2013b*; *Cobey et al., 2018*; *Huang et al., 2015*; *Petrie et al., 2016*). Therefore, the reports from the biannual influenza

vaccine-strain selection meetings have increasingly referenced titer measurements for human sera, made using either individual sera or serum pools (*World Health Organization, 2024b*; *World Health Organization, 2017*; *World Health Organization, 2019*; *World Health Organization, 2020*; *World Health Organization, 2022*).

Here, we use a new sequencing-based assay (*Loes et al., 2024*) to measure the neutralization titers of 78 recent H3N2 viral strains by a large set of children and adult sera. These measurements quantify the heterogeneity of neutralizing antibody immunity to influenza across different members of the population. We find that the evolutionary success of different H3N2 strains is highly correlated with the fraction of sera that have low titers against each strain, suggesting that large-scale sequencing-based neutralization assays can help inform understanding of influenza virus evolution.

## Results

### A sequencing-based assay to measure neutralization titers against H3N2 influenza strains

To measure serum neutralization titers against a large number of human influenza virus HA strains, we utilized a recently developed sequencing-based assay (*Loes et al., 2024*). In this approach, influenza viruses are created, encoding different HA strains, each tagged with an identifying nucleotide barcode (*Welsh et al., 2024*; *Loes et al., 2024*; *Bacsik et al., 2023*; *Figure 1a* and *Figure 1—figure supplement 1*). The barcoded viral variants are then pooled and assayed in an experimental format similar to a traditional neutralization assay (*Figure 1b*) but with the infectivity of each variant quantified by Illumina sequencing of the viral barcodes (*Figure 1c*). This sequencing-based readout enabled us to measure 1872 neutralization curves per 96-well plate (triplicate measurements for 78 viruses against 8 sera).

We designed a virus library in November 2023 with the goal of covering the diversity of H3N2 HA sequences present at the beginning of the 2023 Northern Hemisphere season. For the library, we selected HA haplotypes with high frequencies or a substantial number of descendants within a 12-month window preceding November 2023. This process identified 62 HA haplotypes; for each haplotype, we selected a representative naturally occurring strain HA for inclusion in the library. These strains are widely distributed across the branches of the Nextstrain 2-year H3N2 seasonal influenza phylogenetic tree from late 2023 (*Figure 2a*). We also supplemented the library with the HAs from Northern Hemisphere egg- and cell-passaged vaccine strains from all seasons between 2014 and 2024 (*Figure 2b* and *Supplementary file 1*).

Our library design successfully covered most of the diversity of human H3N2 HA in 2023, although by 2024, there were an increasing number of human H3N2 HA sequences not well represented by a strain in our library (*Figure 2c*, *Figure 2—figure supplement 1a*). The HA protein sequence diversity among the 2023-circulating strains in the library was fairly low, reflecting the modest standing genetic diversity of H3N2 HA in 2023. Library strains were usually separated from their closest relative in the library by just a few HA amino acid mutations (*Figure 2—figure supplement 1b–e*). The greatest variation among the library strains was generally at sites in classically defined antigenic regions on the globular head of the HA protein (*Figure 2—figure supplement 1f and g*).

We generated barcoded viruses for each of the 78 HA sequences we had chosen, making three independently barcoded viruses for each HA sequence (*Figure 1*). Note that all non-HA genes (including neuraminidase) for these viruses are from the lab-adapted A/WSN/1993 (H1N1) strain. To determine the relative titers of all the strains, we initially pooled them at equal volumes, infected cells, and quantified the relative frequency of each strain by barcode sequencing (*Figure 1—figure supplement 1d*). We then re-pooled the strains based on these relative titers to generate a pooled library where each HA was at approximately equal frequencies (*Figure 1—figure supplement 2*).

### Neutralization titers to 2023 strains are heterogeneous across sera from children and adults

We measured neutralization titers of all 78 viral strains against a total of 150 serum samples from children and adults with diverse ages and vaccination histories (*Table 1*), representing 11,700 total titer measurements. The measured titers were highly reproducible both across different barcodes for the same HA and assays performed on different days (*Figure 3—figure supplement 1*).

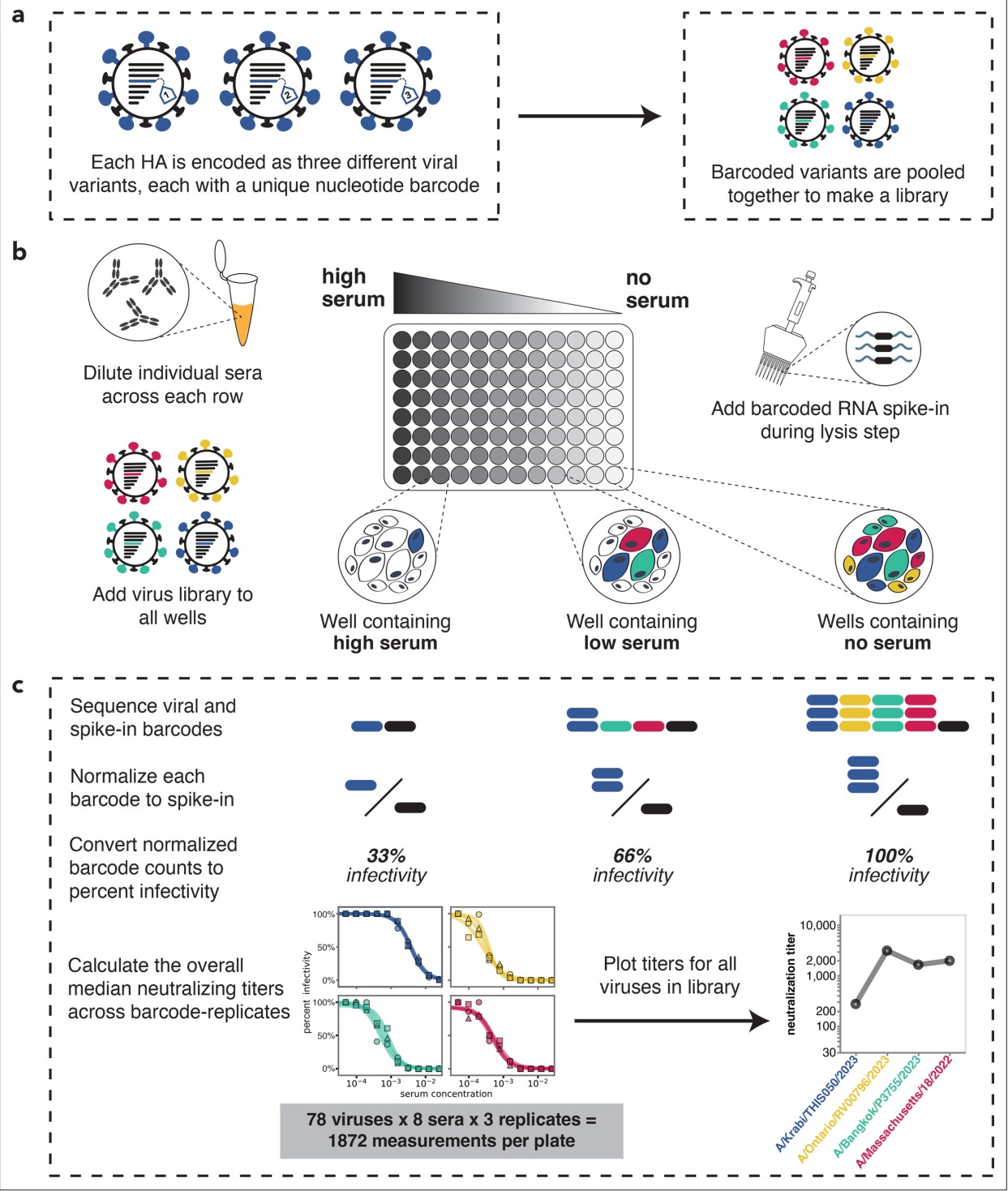

**Figure 1.** Overview of sequencing-based neutralization assay. (**a**) We generate a library of barcoded influenza viruses carrying different hemagglutinins (HAs), each identified by a unique 16-nucleotide barcode in its genome. See *Figure 1—figure supplement 1* for details. (**b**) The virus library is incubated with sera and then added to MDCK-SIAT1 cells in a 96-well plate. At 12–14 hr post-infection, cells are lysed and a known concentration of barcoded RNA spike-in is added to each well. (**c**) The percent infectivity of each viral variant at each serum concentration is calculated by determining the barcode counts by sequencing. Viral barcode counts are normalized by dividing the viral barcode counts by the barcoded RNA spike-in counts. Percent infectivities are calculated by dividing the normalized barcode counts for each serum concentration by those in the no-serum control wells. These percent infectivities are used to fit neutralization curves and determine the neutralization titer, which is defined as the reciprocal of the serum

*Figure 1 continued on next page*

*Figure 1 continued*

dilution at which 50% of the virus is neutralized. Note that since each HA is associated with three different barcodes, each neutralization titer is measured in triplicate; we report the median across the three replicates.

The online version of this article includes the following figure supplement(s) for figure 1:

**Figure supplement 1.** Design of chimeric barcoded hemagglutinin (HA) construct and virus library generation.

**Figure supplement 2.** Determining and validating the cell density and viral library concentration for sequencing-based neutralization assays.

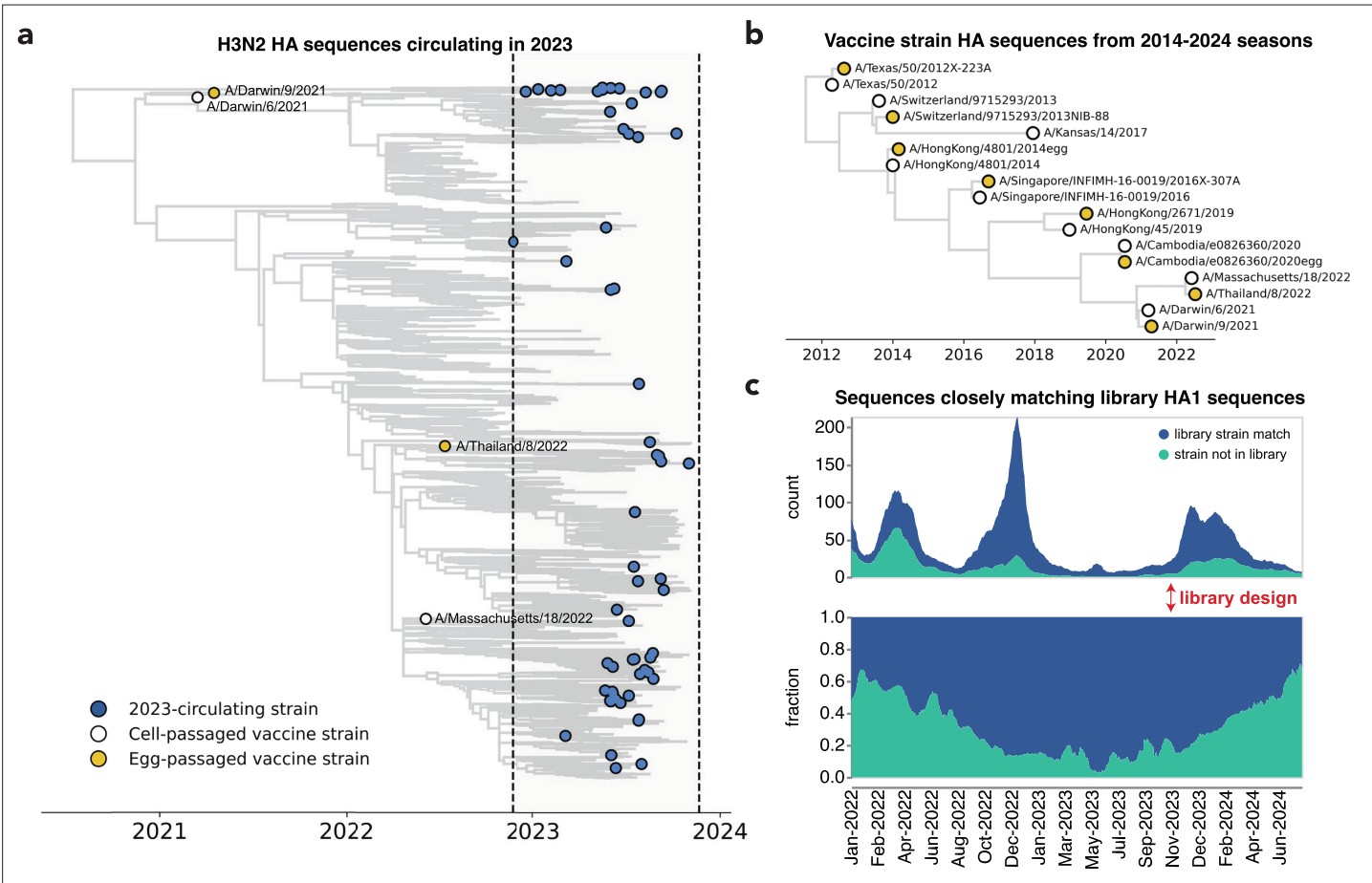

**Figure 2.** The library covers most H3N2 hemagglutinin (HA) diversity in 2023. (**a**) Phylogenetic tree of HAs from recent strains in the library, with 2023-circulating strains in blue, cell-passaged vaccine strains in white and egg-passaged vaccine strains in yellow. The cell-passaged vaccine strain A/Massachusetts/18/2022 is identical at the amino acid level to a strain circulating in 2023 and is therefore also classified as a 2023-circulating strain in our analyses. See https://nextstrain.org/groups/jbloomlab/seqneut/h3n2-ha-2023-2024 for an interactive view of the full library on a background of the 2-year Nextstrain tree available in November 2023. (**b**) Phylogenetic tree of HAs from cell- and egg-passaged strains corresponding to H3N2 component of the Northern Hemisphere influenza vaccine from 2014 to 2024 (*Supplementary file 1*). Cell-passaged vaccine strains are in white and egg-passaged vaccine strains are in yellow. A virus with the HA from the egg-passaged A/Kansas/14/2017 strain could not be grown in our system and was excluded. (**c**) The fraction of all sequenced human H3N2 strains with HAs that closely match HA1 sequences within our library from January 2022 to June 2024. The rolling means (±10 days) of total sequence counts and fraction of sequences are plotted, with strains matched in the library in blue and strains not represented by the library in green. A close match was defined as being within one amino acid mutation in the HA1 domain of HA. The HA1 domain encompasses sites where the majority of antigenic evolution takes place (*Neher et al., 2016*; *Muñoz and Deem, 2005*; *Shih et al., 2007*; *Koel et al., 2013*; *Wilson and Cox, 1990*; *Wiley et al., 1981*), but a similar analysis with full HA ectodomain sequences produces a qualitatively similar result (*Figure 2—figure supplement 1a*).

The online version of this article includes the following figure supplement(s) for figure 2:

**Figure supplement 1.** Hemagglutinin (HA) diversity among the strains in the library.

**Table 1.** Overview of children and adult cohorts.

Summary of sera from the children and adult vaccination cohorts. Children's sera were from routine hospital or clinic visit blood draws and had limited information about vaccine and exposure histories pulled from electronic health records. Adult sera were collected through vaccine response studies based in the United States of America (USA) and Australia at various time points pre-vaccination and post-vaccination. Detailed metadata can be found at https://github.com/jbloomlab/flu_seqneut_H3N2_2023-2024/tree/main/data/sera_metadata.

| | Pediatric | Adult vaccination (egg) | Adult vaccination (cell) |
|---|---|---|---|
| Number of individuals | 56 | 39 | 8 |
| Age range | 1–14 years | 22–74 years | 26–54 years |
| Location | Seattle (USA) | Philadelphia (USA) | Australia |
| Sex distribution | 26 F/30 M | 24 F/15 M | 6 F/2 M |
| Recent vaccination type | Unknown | Northern Hemisphere 2023–2024 egg-based vaccine (A/Darwin/9/2021) | Southern Hemisphere 2024 cell-based vaccine (A/Massachusetts/18/2022) |
| Matched samples? | No | Yes, at 0 and 28 days post-vax | Yes, 0 and 17–21 days post-vax |
| Date samples collected | Dec-23 | Oct-Dec 2023 | Apr-May 2024 |

We first focus our analysis on the titers for sera collected in the second half of 2023 from 56 children (ages 1–14 years) and 39 pre-vaccination adults (ages 22–74 years). Most of these adults had been vaccinated in the prior year, while the vaccination histories of the children remain mostly unknown. All these sera were collected from locations in the USA (*Table 1*). We reasoned that these sera likely provide a reasonable representation of population neutralizing antibody immunity at the beginning of the 2023–2024 Northern Hemisphere influenza season.

There was substantial variation in neutralization titers across sera, and in many cases also across different viral strains for the same serum. For example, *Figure 3a* shows neutralization titers for a child and adult serum sample against the 62 viruses that represented HA diversity in 2023. Each serum has reduced titers to certain groups of strains (*Figure 3a*). The child has reduced titers to strains with a mutation at site 145, while the adult is unaffected by mutations at site 145 but has decreased titers to strains with mutations at sites 275 and 276 (*Figure 3a*). Sites 145, 275, and 276 are all in known antigenic regions of HA (*Muñoz and Deem, 2005*). Notably, mutations at sites 145 and 276 both subsequently increased in frequency among human H3N2 in 2024 (*Figure 3—figure supplement 2a–c*), and the strain chosen for the 2024 Southern Hemisphere vaccine contains mutations at both these sites (*World Health Organization, 2024a*).

Some of the heterogeneity apparent in these two example neutralization profiles (*Figure 3a*) is mirrored across the larger set of 56 children and 39 adult sera (*Figure 3b*). The most striking observation is the wide person-to-person variation in titers, which is especially apparent for the children sera (*Figure 3—figure supplement 2d and f*). For instance, a few children's sera neutralize all strains with titers >1000, but most children's sera titers are roughly an order of magnitude lower (*Figure 3b*). There is also person-to-person variation in titers across adult sera, although less so than for the children sera (*Figure 3b*, *Figure 3—figure supplement 2d–f*). Nested within this person-to-person variation is strain-to-strain variation in titers for each serum. While the distribution of within-serum, strain-to-strain variation is not substantially different between age cohorts (*Figure 3—figure supplement 2g*), for a few specific strains, the strain-to-strain variation does segregate by age cohort. For instance, across the entire sera set, strains with mutations at site 145 have relatively lower titers than other strains for the children sera, but not for the adult sera (*Figure 3*). However, much of the person-to-person and strain-to-strain variation seems idiosyncratic to individual sera rather than age groups, since the

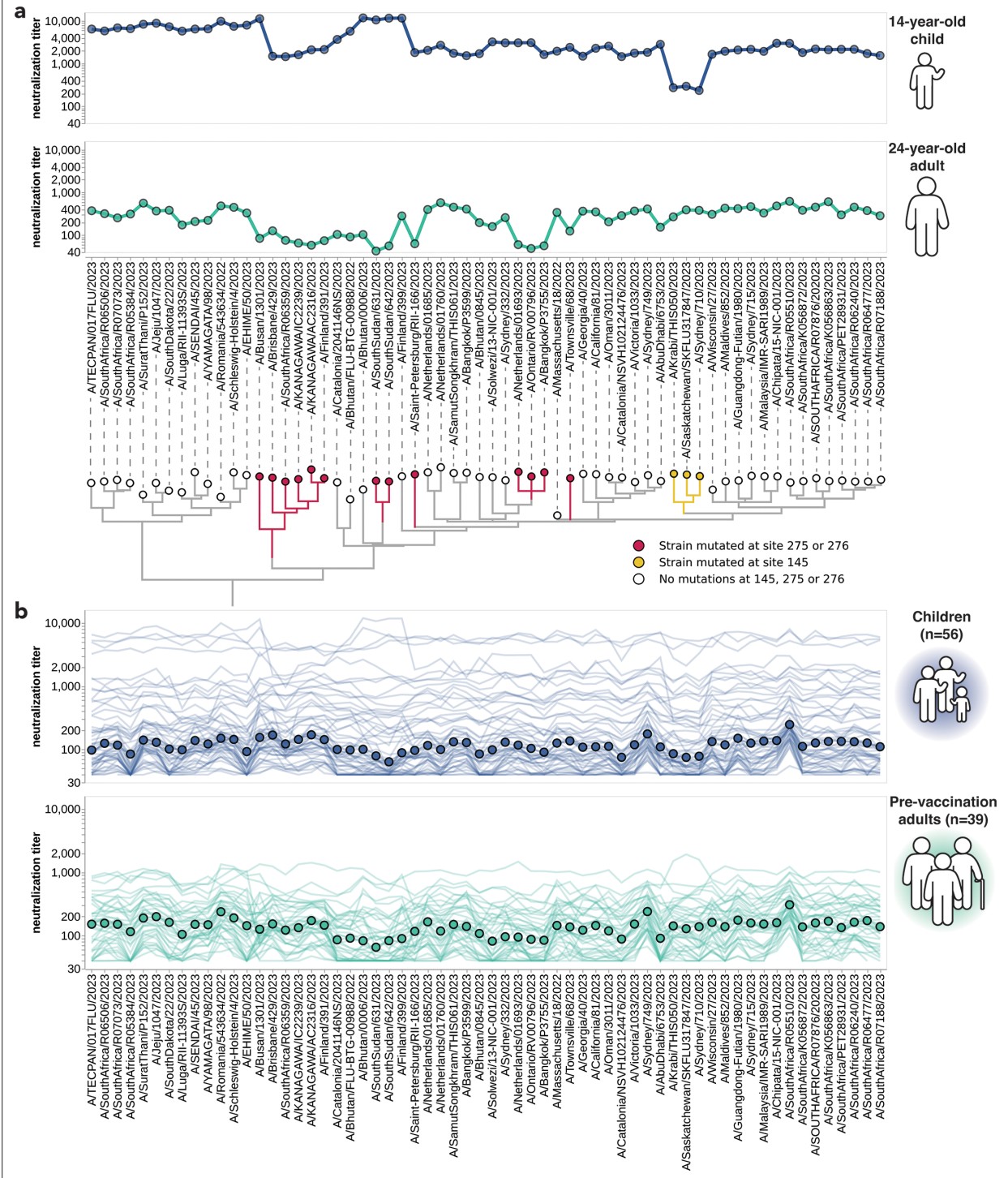

**Figure 3.** Neutralization titers to 2023-circulating strains for children and adults vary among individuals and cohorts. (**a**) Neutralization titer profiles for a child and a pre-vaccination adult, showing the titer of each serum against each of the 2023-circulating strains in the library. Neutralization by the child's serum is reduced for strains with the S145N mutation in antigenic region A, while neutralization by the adult's serum is reduced by multiple mutations within antigenic region C (sites 275 and 276). Strains are grouped phylogenetically on the x-axis. (**b**) Neutralization titer profiles across all individuals from the children and adult pre-vaccination cohorts. Each thin line is a neutralization titer profile for an individual serum. Each point represents the median neutralization titer across all sera for that strain.

The online version of this article includes the following figure supplement(s) for figure 3:

**Figure supplement 1.** Within- and between-plate titer measurements are highly correlated.

*Figure 3 continued on next page*

*Figure 3 continued*

**Figure supplement 2.** H3 hemagglutinin (HA) phylodynamics, birth year cohorts, and age cohorts explain some patterns in neutralization titers.

**Figure supplement 3.** Fold change in titer for each viral strain relative to the median titer across all strains for each children and pre-vaccination adult sera.

**Figure supplement 4.** Neutralizing titer correlations and their relationship with age difference between pairs of sera and Hamming distance between pairs of viruses.

**Figure supplement 5.** Neutralizing titers across all vaccine and 2023-circulating strains plotted as a heatmap.

heterogeneity persists even if the sera are analyzed in more fine-grained age groups (*Figure 3— figure supplement 2h*).

## Pooled sera fail to capture the heterogeneity of individual sera

One approach that has been used for influenza antigenic characterization is to pool sera from many different individuals and then measure titers against the serum pool (*World Health Organization, 2017*; *World Health Organization, 2019*; *World Health Organization, 2020*; *World Health Organization, 2022*). However, given the dramatic heterogeneity across sera (*Figure 3*), these pools might be expected to mostly just reflect the properties of the most potent sera in the pool rather than capturing the full heterogeneity.

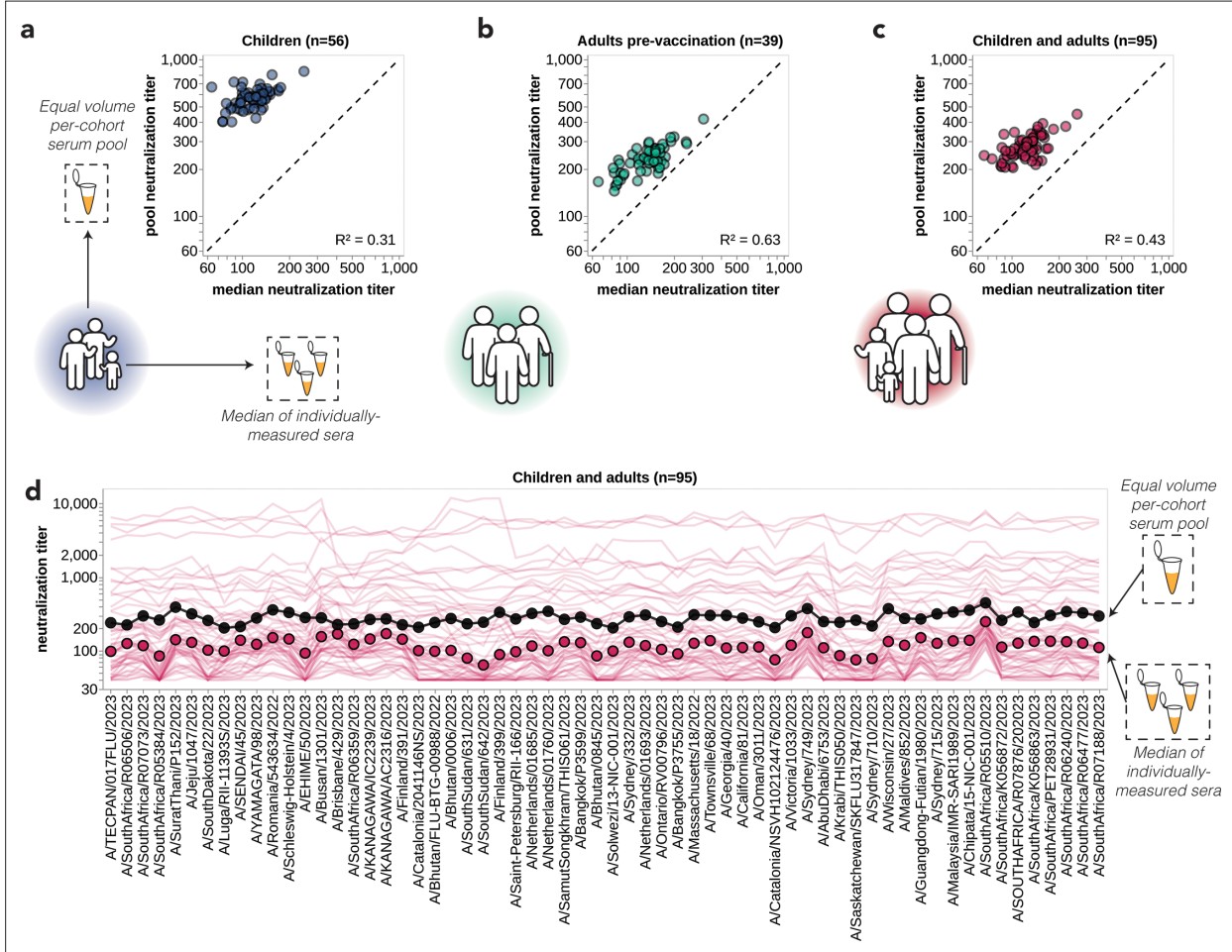

**Figure 4.** Pooled sera neutralization titer profiles do not capture nuances from individually measured sera. Correlations between titers measured from pooled sera and the median of individually measured titers are plotted for (**a**) children, (**b**) pre-vaccination adults, and (**c**) children and pre-vaccination adults together. Each dot corresponds to the pooled or median titer for a given 2023-circulating library strain. (**d**) The full neutralization profiles for all children and pre-vaccination adults individually and as a serum pool, replotted from *Figure 3b*. Titers for individual sera are plotted as thin red lines, with the median titer across all children and adult sera indicated by red points, and the titer of the pooled serum indicated by a black line and points.

To confirm that sera pools are not reflective of the full heterogeneity of their constituent sera, we created equal volume pools of the children and adult sera and measured the titers of these pools using the sequencing-based neutralization assay. As expected, neutralization titers of the pooled sera were always higher than the median across the individual constituent sera, and the pool titers against different viral strains were only modestly correlated with the median titers across individual sera (*Figure 4*). The differences in titers across strains were also compressed in the serum pools relative to the median across individual sera (*Figure 4*). The failure of the serum pools to capture the median titers of all the individual sera is especially dramatic for the children sera (*Figure 4*) because these sera are so heterogeneous in their individual titers (*Figure 3b*). Taken together, these results show that serum pools do not fully represent individual-level heterogeneity and are similar to taking the arithmetic mean of the titers for a pool of individuals, which tends to be biased by the highest titer sera.

## Evolutionary success of viral strains correlates with neutralization titers from individually measured sera but not pooled sera

The most evolutionarily successful viral strains each human influenza season are those that jointly maximize both inherent transmissibility and evasion of pre-existing population immunity (*Smith et al., 2004*; *Neher et al., 2016*; *Kim et al., 2024*; *Luksza and Lässig, 2014*; *Huddleston et al., 2020*; *Strelkowa and Lässig, 2012*). Several studies have shown that experimental measurements of anti-HA antibody immunity (e.g. hemagglutination inhibition or neutralization assays) can partially explain the evolutionary competition among human influenza strains, presumably by capturing some of the contribution of immune evasion to viral fitness (*Smith et al., 2004*; *Neher et al., 2016*; *Kim et al., 2024*; *Luksza and Lässig, 2014*; *Huddleston et al., 2020*; *Strelkowa and Lässig, 2012*). Our sequencing-based neutralization assay enabled us to measure human neutralizing antibody titers to a wide range of influenza strains on an unprecedented scale, so we sought to test whether these measurements correlated with the evolutionary success of these strains.

To estimate the evolutionary success of different human H3N2 influenza strains during 2023, we used multinomial logistic regression, which uses sequence counts to estimate fixed strain growth rates relative to a baseline strain for the entire analysis time period (in this case, 2023) (*Abousamra et al., 2024*; *Annavajhala et al., 2021*; *Obermeyer et al., 2022*). Relative growth rates estimated by multinomial logistic regression represent relative fitnesses of strains over that time period. There were sufficient sequencing counts to reliably estimate growth rates in 2023 for 12 of the HAs for which we measured titers using our sequencing-based neutralization assay libraries (*Figure 5a, b*, *Figure 5—figure supplement 1*). We estimated strain growth rates relative to the baseline strain of A/Massachusetts/18/2022. Note that these growth rates estimate how rapidly each strain grows relative to the baseline strain, rather than the absolute highest frequency reached by each strain. Each strain's absolute growth rate corresponds to the slope of the strain's logit-transformed frequencies at the end of the analysis time period (*Figure 5—figure supplement 1*).

We compared the estimated strain-specific growth rates to the measured neutralization titers across the entire set of children and adult sera for each strain. Susceptibility to influenza is thought to increase once neutralization titers fall below a threshold (*Hobson et al., 1972*; *Kim et al., 2024*; *Petrie et al., 2016*), so we correlated the growth rates to the fraction of sera with titers below a threshold against each strain, testing a range of thresholds. The growth rates were most correlated with the fraction of sera with neutralization titers below 138 (*Figure 5c*), and this correlation was highly statistically significant as assessed by the fact that it exceeded the correlation for 200 different random permutations of the experimental titer data (*Figure 5d*). This finding demonstrates that the evolutionary success of human H3N2 influenza strains in 2023 was highly correlated with experimental measurements of the strains' ability to evade neutralizing antibody immunity across the human population. Similar correlations are observed if we quantify the per-strain neutralization titers in terms of the geometric mean or median of the neutralization titers rather than by the fraction of sera below a titer threshold (*Figure 5—figure supplement 2a*).

The above correlations were for growth rates estimated for strains with at least 80 sequencing counts in 2023. We also repeated the above analysis using lower cutoffs for how many sequencing counts are needed to estimate a strain's growth rate; these estimates include more strains but are noisier. For these lower cutoffs, there remained a good although somewhat weaker correlation

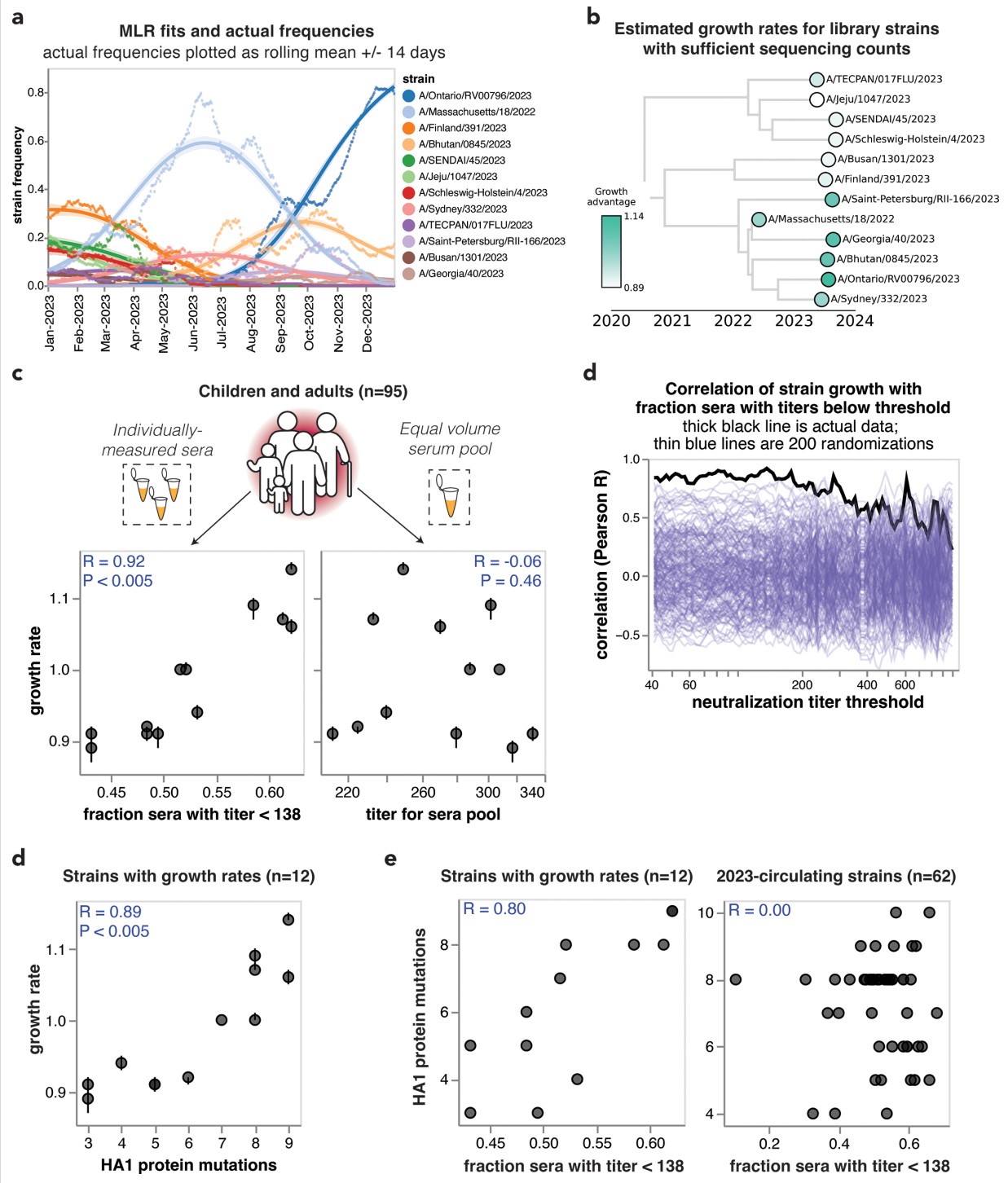

**Figure 5.** Individually measured serum neutralization titers correlate with the growth rate of viral strains in the human population in 2023. (**a**) Strain frequencies and model fits for the strains with sufficient sequencing counts to estimate a growth rate using multinomial logistic regression. Dots represent strain frequencies averaged across a 14-day sliding window, and lines represent the model fit. (**b**) Phylogenetic tree of the 12 viral strains with sufficient sequencing data to estimate their relative growth rates. Each strain is colored by its growth rate relative to the A/Massachusetts/18/2022 strain. (**c**) Correlation between the strain growth rates and serum neutralization titers from 95 children and pre-vaccination adults. The plot on the left shows the fraction of sera with titers below 138. The plot at right shows the titers of an equal volume pool of all sera. In both panels, each point corresponds to the growth rate and titer for one of the 12 viral strains. The numbers in the upper corners show the Pearson correlation coefficient R and the p-value as assessed by randomizing the experimental data among strains 200 times. (**c**) Correlation of strain growth rate with fraction of sera that have a titer below a given threshold for thresholds between 40 and 1000. The thick black line shows the actual correlation at each threshold, and the thin blue

*Figure 5 continued*

lines show the correlations for 200 randomizations of the experimental data among the strains. The threshold that gave the highest correlation is 138; none of the randomizations had a correlation as high as the actual data at any threshold, so the p-value is <0.005. (**d**) Correlation of strain growth rate and the number of HA1 amino acid mutations relative to the common ancestor of the 12 strains with growth rate estimates. Correlations are similar in strength for full HA ectodomain amino acid mutations and HA nucleotide mutations (***Figure 5—figure supplement 2***). (**e**) Correlations between the fraction of sera with titers below 138 and the number of HA1 amino acid mutations for the 12 strains with estimated growth rates (left) and all 62 of the 2023-circulating strains in the library (right).

The online version of this article includes the following figure supplement(s) for figure 5:

**Figure supplement 1.** Multinomial logistic regression model fits of strain growth rates.

**Figure supplement 2.** Additional growth rate comparisons with neutralization titers and evolutionary distances.

between growth rate and the fraction of sera below a titer threshold (***Figure 5—figure supplement 2b***).

However, there was no correlation between the growth rates of the strains and their neutralization titers against the pooled children and adult sera (***Figure 5c***), emphasizing the importance of accounting for population heterogeneity. Broadly similar trends held if the analyses were done using neutralization titers for only the children or only the adult sera. Specifically, there were strong correlations between strain growth rate and fraction of sera with titers below a threshold against each strain when children and adult sera were analyzed individually, although the correlations were not quite as strong as for the combined children and adult sera set (***Figure 5—figure supplement 2c–h***). But growth rates were at best modestly correlated with titers against pools of only children or only adult sera (***Figure 5—figure supplement 2c–h***). The reduced disparity between the full sera set versus serum pool correlations for the children-only and adult-only sera sets compared to the full combined sera set could be because the reduced heterogeneity within the children-only and adult-only sets makes the pools a better (although still imperfect) proxy for these sera sets versus the full combined sera set (***Figure 4a–c***, ***Figure 5—figure supplement 2i***).

The strong correlation between strain growth and the fraction of sera with titers below a threshold is highly significant as assessed by randomizing the experimental data among strains (***Figure 5c and d***). However, an important caveat is that the strains are phylogenetically related and growth rate may segregate with phylogeny (***Figure 5b***), and some strains share HA1 mutations due to common descent (***Table 2***). Accurately assessing the significance of relationships involving variables with

**Table 2.** HA1 mutations present in strains with estimated growth rates.

HA1 amino acid mutations for each of the 12 strains with sufficient sequence counts to estimate growth rates. Mutations in known antigenic regions as defined by ***Muñoz and Deem, 2005***, are in bold text, and mutations outside of antigenic regions are in regular weight text. The table lists all HA1 mutations relative to the most-recent common ancestor of the 12 strains.

| Strain name | HA1 substitutions |
|---|---|
| A/TECPAN/017FLU/2023 | **E50K, F79V, I140K**, I242M |
| A/Jeju/1047/2023 | **E50K, F79V, I140K** |
| A/SENDAI/45/2023 | **E50K, F79V, T135A, I140K, S262N** |
| A/Schleswig-Holstein/4/2023 | R33Q, **E50K, F79V, I140K, S262N** |
| A/Busan/1301/2023 | **D53G, H156S, K276R** |
| A/Finland/391/2023 | **D53G**, D104G, **I140K, H156S, K276R**, R299K |
| A/Saint-Petersburg/RII-166/2023 | **E50K, D53N**, N96S, **I140K, H156S, Q173R**, I192F, I223V, **K276E** |
| A/Massachusetts/18/2022 | **E50K, D53N**, N96S, **I140K, H156S**, I192F, I223V |
| A/Georgia/40/2023 | **E50K, D53N**, N96S, **I140K, H156S**, I192F, I223V, I242M |
| A/Bhutan/0845/2023 | I25V, **E50K, D53N**, N96S, **I140K, H156S**, I192F, I223V |
| A/Ontario/RV00796/2023 | **E50K, D53N**, N96S, N122D, **I140K, H156S**, I192F, I223V, **K276E** |
| A/Sydney/332/2023 | **E50K, D53N**, N96S, N122D, **I140K, H156S**, I192F, I223V |

phylogenetic structure is notoriously challenging from a statistical perspective (*Felsenstein, 1985*). To assess whether phylogenetic structure may contribute, we examined the correlation of strain growth and the number of HA1 amino acid mutations relative to the common ancestor of all strains with growth estimates. This correlation was also strong and significant (*Figure 5d*, *Figure 5—figure supplement 2j*), an expected result because the neutralization titers correlate with the number of HA1 mutations for the strains with sufficient sequences to estimate growth rates, although they do not correlate for the set of all strains (*Figure 5e*). A permutation feature importance analysis in the context of multiple linear regression indicated that the fraction of individuals with low neutralization titers is more important for explaining strain growth than the HA1 mutation count (68% importance for titers versus 32% for HA1 mutation counts), but the variables are so co-linear that they cannot be convincingly separated. Therefore, with the current data, it is not possible to be confident that the experimentally measured titers provide substantially more information about the evolutionary success of viral strains than simple phylogenetic methods such as counting mutations.

## Patterns of neutralization of the past decade of vaccine strains stratify largely by birth cohort

While the preceding analyses demonstrated the utility of sequencing-based neutralization assays for measuring titers of currently circulating strains, our library also included viruses with HAs from each of the H3N2 influenza Northern Hemisphere vaccine strains from the last decade (2014–2024, see *Supplementary file 1*). These historical vaccine strains cover a much wider span of evolutionary diversity than the 2023-circulating strains analyzed in the preceding sections (*Figure 2a, b*, *Figure 2—figure supplement 1b–e*). For this analysis, we focused on the cell-passaged strains for each vaccine, as these are more antigenically similar to their contemporary circulating strains than the egg-passaged

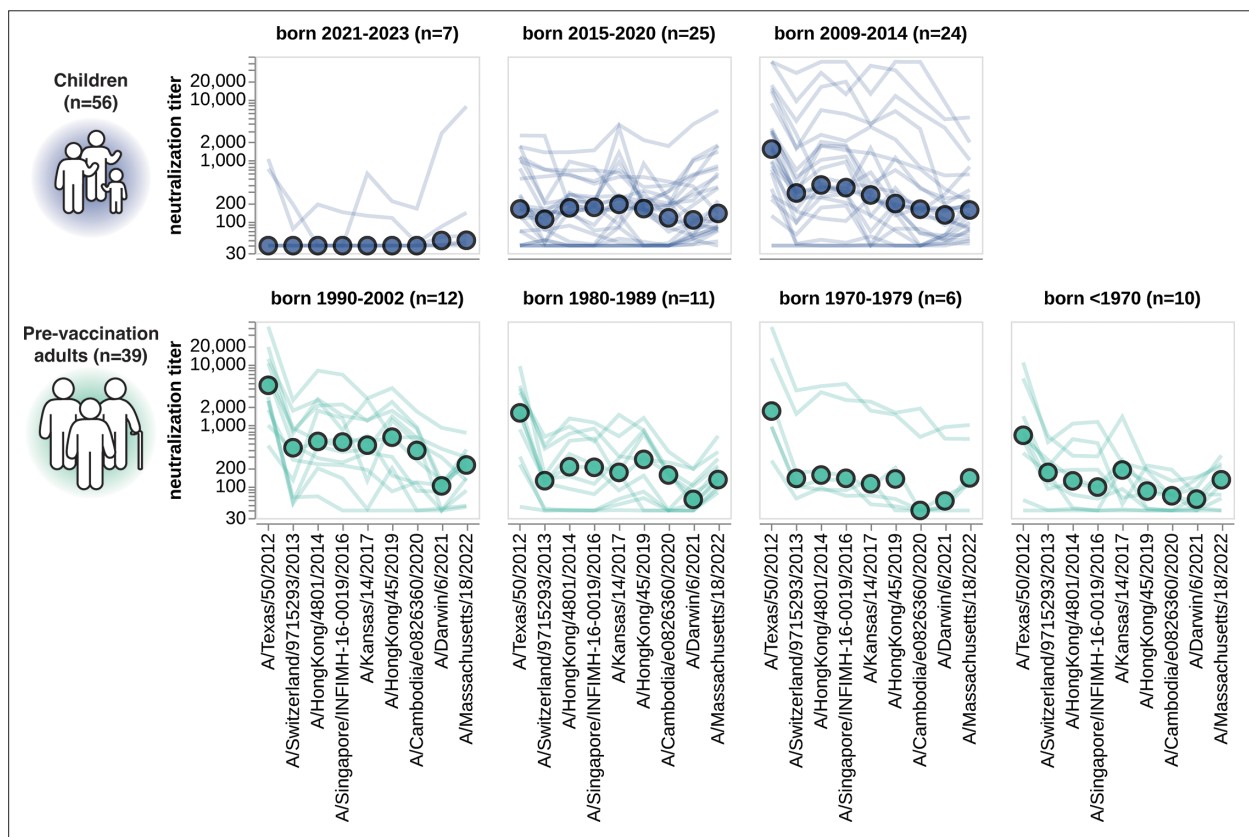

**Figure 6.** Neutralization titers to the past decade of vaccine strains. Neutralization titer profiles to the past decade of cell-passaged Northern Hemisphere vaccine strains across all individuals from the children and adult pre-vaccination cohorts, stratified by age group as indicated in plot panel titles. Strains are ordered on the x-axis by the year they were collected. Each thin line is a neutralization titer profile for an individual serum, and each colored point represents the median titer across all sera for that strain. See *Supplementary file 1* for details on which strains were in the vaccines for which seasons.

vaccine strains since they lack the mutations that arise during growth of viruses in eggs (*Zost et al., 2017*; *Guthmiller et al., 2021*; *Wu et al., 2019*; *Supplementary file 1*).

The patterns of neutralization of the vaccine strains can mostly be rationalized in terms of exposure history. Children born between 2021 and 2023 generally have either no measurable titer to any strain or their highest titers to recent vaccine strains (*Figure 6*), presumably because these young children were either not yet exposed to H3N2 influenza or were exposed to a recent strain. Children born between 2015 and 2020 often neutralize the past decade of vaccine strains (*Figure 6*), consistent with the fact that children of this age have typically been exposed to H3N2 influenza (*Ranjeva et al., 2019*; *Kucharski et al., 2018*). However, the strain that is best neutralized by children born between 2015 and 2020 varies, potentially because different children in this age group were first infected by different viruses over the last decade. Older children born between 2009 and 2014 typically best neutralize the oldest vaccine strain tested (*Figure 6*), consistent with numerous studies showing that an individual's neutralizing titers are highest to the influenza strains to which they were first exposed (*Ranjeva et al., 2019*; *Francis, 1960*; *Lessler et al., 2012*; *Fonville et al., 2014*; *Kucharski et al., 2018*; *Yang et al., 2020*; *Davenport et al., 1953*). Adults across age groups also have their highest titers to the oldest vaccine strain tested (*Figure 6*), consistent with the fact that these adults were likely first imprinted by exposure to an older strain more antigenically similar to A/Texas/50/2012 (the oldest strain tested here) than more recent strains. Note that a similar trend toward adult sera having higher titers to older vaccine strains was also observed in a more recent study we have performed using the same methodology described here (*Kikawa et al., 2025*).

## Vaccination of adults broadly increases neutralization titers to most strains

Our sequencing-based assay could also be used to assess the impact of vaccination on neutralization titers against the full set of strains in our H3N2 library. To do this, we analyzed matched 28-day post-vaccination samples for each of the above-described 39 pre-vaccination samples from the cohort of adults based in the USA (*Table 1*). We also analyzed a smaller set of matched pre- and post-vaccination sera samples from a cohort of eight adults based in Australia (*Table 1*). Note that there are several differences between these cohorts: the USA-based cohort received the 2023–2024 Northern Hemisphere egg-grown vaccine whereas the Australia-based cohort received the 2024 Southern Hemisphere cell-grown vaccine, and most individuals in the USA-based cohort had also been vaccinated in the prior season whereas most individuals in the Australia-based cohort had not. Therefore, multiple factors could contribute to observed differences in vaccine response between the cohorts.

For both cohorts, vaccination broadly increased neutralization titers to all 2023-circulating strains. For the USA-based cohort, vaccination typically increased titers to the 2023-circulating strains by ~2- to 4-fold whereas for the Australia-based cohort the increases were typically ~5- to 10-fold (*Figure 7a*). Due to the multiple differences between cohorts, we are unable to confidently ascribe a cause to these differences in magnitude of vaccine response.

Vaccination also generally increased titers to the last decade of both cell- and egg-passaged vaccine strains, although the increases tended to be largest to the most recent vaccine strains (*Figure 7b*). We therefore see modest evidence for the previously described phenomenon of back-boosting (*Fonville et al., 2014*), where vaccination with a recent strain increases titers to both recent and older strains. However, the extent of back-boosting is quite small for the oldest vaccine strains. We also found much higher titers against egg-passaged strains relative to the cell-passaged strains of the same year in both the USA-based and Australia-based cohorts, both pre- and post-vaccination (*Figure 7b*). The higher titers to the egg-passaged versus cell-passaged vaccine strains could be due to antigenic differences between the strains, or due to differences in receptor avidity that could make egg-passaged strains inherently easier to neutralize (*Hensley et al., 2009*; *Li et al., 2013a*; *Kodihalli et al., 1995*; *Ito et al., 1997*; *Gambaryan et al., 1999*).

## Discussion

We have used a sequencing-based assay (*Loes et al., 2024*) to experimentally measure neutralization titers for sera from humans of different ages against a large set of H3N2 viral strains. There was substantial person-to-person variation in titers to different H3N2 strains that circulated in 2023. The

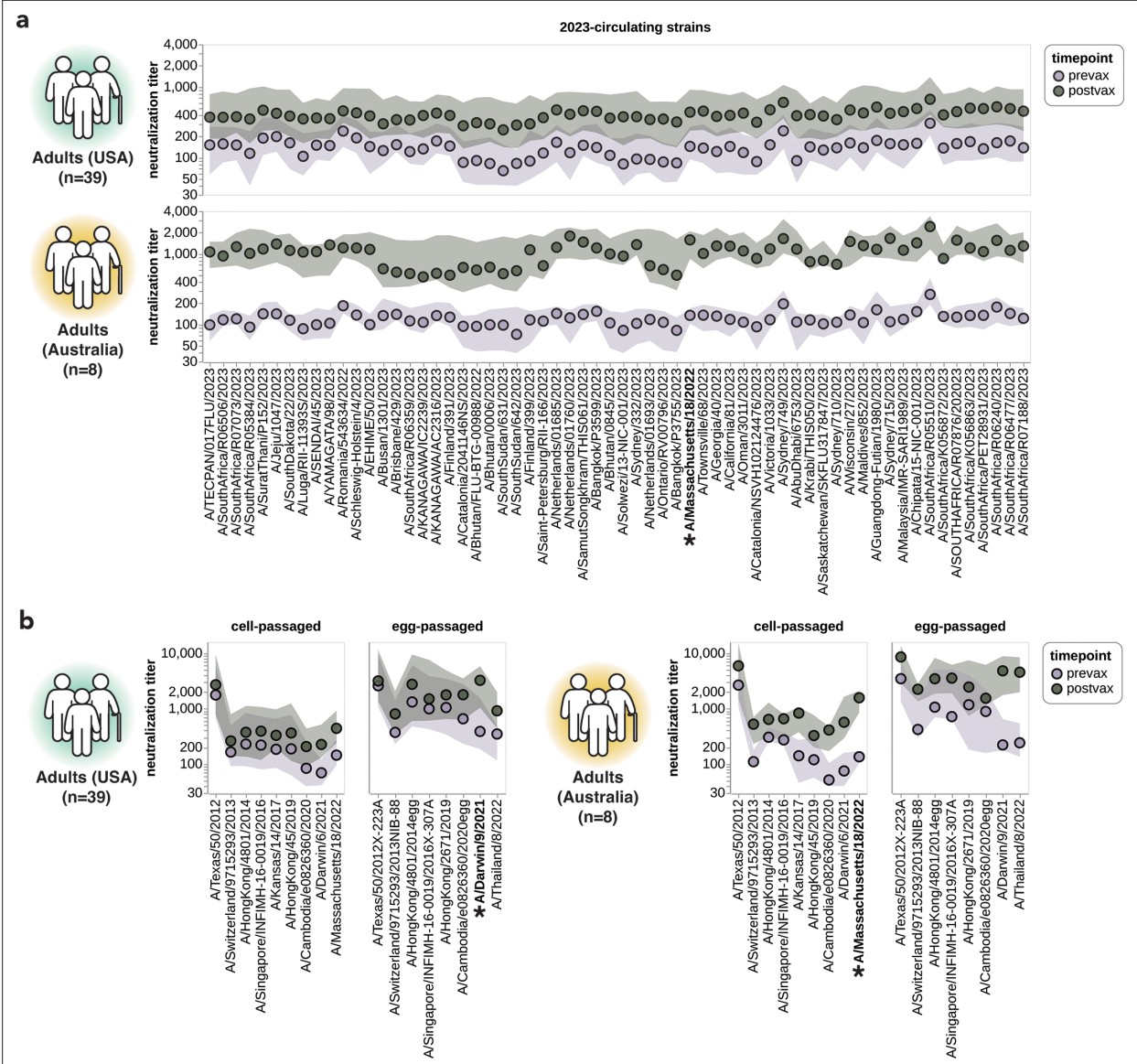

**Figure 7.** Impact of vaccination on neutralization titers. (**a**) Neutralization titers pre- and post-vaccination for the USA-based adult cohort (top) and the Australia-based adult cohort (bottom) against the 2023-circulating library strains. Points indicate the median titers across participants, and the shaded regions show the interquartile range. The H3N2 component strain of the 2024 Southern Hemisphere seasonal influenza vaccine (A/Massachusetts/18/2022) is indicated by bold text and an asterisk; this strain circulated in 2023 and so was classified as both a 2023-circulating strain and a vaccine strain in our analysis. (**b**) Neutralization titers pre- and post-vaccination for the two cohorts to each cell-passaged or egg-passaged Northern Hemisphere H3N2 vaccine strain from the past decade (*Supplementary file 1*). Viruses are listed on the x-axis in order of increasing year in which they were isolated. The USA-based cohort received the 2023–2024 Northern Hemisphere seasonal influenza vaccine (A/Darwin/9/2021), and the Australia-based cohort received the 2024 Southern Hemisphere seasonal influenza vaccine (A/Massachusetts/18/2022).

highest neutralization titers were for a subset of the sera from children, which in some cases could neutralize all 2023-circulating H3N2 strains over an order of magnitude better than the median titer across all sera. The high titers of some children sera are consistent with the idea that neutralizing antibody responses are highest to strains encountered in the first decades of life (*Lessler et al., 2012*; *Fonville et al., 2014*), although we also note that children are more frequently infected (*Worby et al., 2015*) and could therefore be more likely to have recent immunologic boosting (*Horsfall and Rickard, 1941*; *Arevalo et al., 2020*). However, there were also many children with low to moderate titers, and overall titers were variable both within and between age groups.

We identified many examples of individual sera that had lower titers to specific 2023-circulating H3N2 strains, including strains with mutations at key sites of HA evolution in 2023–2024, such as 145 and 276. However, although there is clear heterogeneity among individuals in their neutralization of 2023-circulating H3N2 strains, only a modest fraction of this heterogeneity can be explained by birth cohort. Therefore, although birth year certainly contributes to strain-to-strain heterogeneity in viral neutralization (*Ranjeva et al., 2019*; *Arevalo et al., 2020*; *Welsh et al., 2024*), among recent strains, there is heterogeneity both between and within birth cohorts. Age does better explain the variation in titers across the past decade of vaccine strains, with older individuals typically better neutralizing older strains, consistent with prior work (*Hennessy et al., 1955*; *Lessler et al., 2012*; *Davenport et al., 1953*; *Francis, 1960*). Therefore, our results suggest that while age explains much of the person-to-person variation in neutralization of highly diverse strains spanning a decade or more of viral evolution, the more fine-grained variation in neutralization of the strains present in a single season can be highly idiosyncratic even within an age group.

The actual evolutionary success (growth rates) of different H3N2 influenza strains in 2023 was highly correlated with the fraction of adult and children sera with low titers to each strain. This finding supports the idea (*Kim et al., 2024*) that evolutionarily successful strains are those that evade the pre-existing immunity of the largest fraction of the human population. In our study, we simply correlated strain growth rates with the titers of a large set of children and adult sera, but future work could try to weight sera to capture the relative importance of age groups in shaping viral transmission at the population level (*Kim et al., 2024*; *Welsh et al., 2024*; *Arevalo et al., 2020*; *Nakajima et al., 2000*; *Sato et al., 2004*; *Worby et al., 2015*). Notably, the neutralization titers of serum pools correlated poorly both with the median titers of the individually measured constituent sera and with the evolutionary success of different viral strains. Therefore, serum pools may be a poor proxy for the heterogeneous population immunity that shapes influenza evolution.

High-throughput sequencing-based neutralization assays like the ones described in this paper could be performed on a timeline that would help inform influenza vaccine-strain selection. Vaccine strains are selected at bi-annual meetings based on phenotypic and antigenic characterization of viruses circulating around the time of each meeting (*Tosh et al., 2010*). Our 78-strain H3N2 influenza library was designed and created in ~12 weeks. With this library in hand, a single graduate student (the lead author of this paper) made ~10,000 neutralization measurements with about 4 cumulative weeks of experimental work. For this initial study, the subsequent data analysis took longer than the experiments, but that analysis could be greatly streamlined now that the computational pipelines are in place. We therefore suggest that libraries designed in the Northern Hemisphere fall could be used to generate data for strain selection in the following spring. The resulting data could help improve vaccine efficacy, which has sometimes been low in part due to antigenic mismatch between selected vaccine strains and circulating strains (*de Jong et al., 2000*; *Xie et al., 2015*). For instance, our experiments identified J.2 subclade strains such as A/Ontario/RV00796/2023 as among those that escaped the largest fraction of children and adults; such strains would have been a good antigenic match in the 2023–2024 season when the J.2 subclade predominated (*World Health Organization, 2024a*). Therefore, sequencing-based neutralization measurements could provide valuable new data that could be incorporated into influenza forecasting models (*Neher et al., 2016*; *Luksza and Lässig, 2014*; *Huddleston et al., 2020*; *Lässig et al., 2017*; *Steinbrück et al., 2014*; *Morris et al., 2018*; *Gandon et al., 2016*) that also include other factors that shape viral evolution.

## Limitations of the study

While our study analyzed sera from a large number of different individuals, these sera were from just a few sources: residual children's sera from a hospital in Seattle and adult sera from heavily vaccinated cohorts in Philadelphia and Australia. While H3N2 influenza strains often circulate globally (*Russell et al., 2008*; *Bedford et al., 2015*), there can be modest geographic stratification of strains that could lead to sera from these locations not being fully representative of global population immunity. And while natural infection plays a stronger role in shaping immunity than vaccination (*Yu et al., 2008*; *Davis et al., 2020*), there could be some differences between the highly vaccinated adults in our study and the unvaccinated population that is more common globally. Additionally, some age groups are lacking from our study, and the number of sera from individuals of different ages may not be proportional to their importance in shaping influenza virus evolution.

Our experiments only measured neutralizing antibodies targeted to HA. While such antibodies are the best-established correlate of protection (*Hobson et al., 1972*; *Tsang et al., 2014*; *Fox et al., 1982*; *Coudeville et al., 2010*; *Ng et al., 2013*; *Ohmit et al., 2011*), non-neutralizing antibodies to HA and anti-neuraminidase antibodies (which are generally not neutralizing in single-round infections like the ones used in our assay; *Liu et al., 1995*) can also contribute to protection (*Krammer, 2019*; *Monto et al., 2015*).

Our comparisons of the neutralization titers to the growth rates of different H3N2 strains were limited by the fact that only a modest number of strains had adequate sequence data to estimate their growth rates. Strains with more sequencing counts tend to be those with moderate-to-high fitness, which therefore limited the dynamic range of growth rates across strains we were able to analyze. Relatedly, the multinomial logistic regression infers a single fixed growth rate per strain for the entire analysis time period of 2023 and cannot represent changes in relative fitness of strains over that relatively short time period. Additionally, because the strains for which we estimated growth rates are phylogenetically related, it is difficult to assess the statistical significance of the correlation (*Felsenstein, 1985*), so it will be important for future work to reassess the correlations with new neutralization data against the dominant strains in future years.

## Methods

### Data and code availability

All data and code are publicly available at the following links:

- Analysis of the sequencing-based neutralization assays: https://github.com/jbloomlab/flu_seqneut_H3N2_2023-2024 (copy archived at *Kikawa and Bloom, 2026*).
- Analyses comparing neutralization titers and growth rates: https://github.com/jbloomlab/flu_H3_2023_seqneut_vs_growth (copy archived at *Bloom and Kikawa, 2026*).
- Computational pipeline for analyzing sequencing-based neutralization assays: https://github.com/jbloomlab/seqneut-pipeline (version 3.1.3) (*Bloom et al., 2024*).
- All measured neutralization titers in CSV format:
  - For the children cohort: https://github.com/jbloomlab/flu_seqneut_H3N2_2023-2024/blob/main/results/aggregated_titers/titers_SCH.csv
  - For the Pennsylvania-based adult vaccine cohort: https://github.com/jbloomlab/flu_seqneut_H3N2_2023-2024/blob/main/results/aggregated_titers/titers_PennVaccineCohort.csv
  - For the Australia-based adult vaccine cohort: https://github.com/jbloomlab/flu_seqneut_H3N2_2023-2024/blob/main/results/aggregated_titers/titers_AusVaccineCohort.csv
  - For the serum pools: https://github.com/jbloomlab/flu_seqneut_H3N2_2023-2024/blob/main/results/aggregated_titers/titers_PooledSera.csv
- Interactive visualizations of all neutralization titers: https://jbloomlab.github.io/flu_seqneut_H3N2_2023-2024/
- The metadata for all tested sera: https://github.com/jbloomlab/flu_seqneut_H3N2_2023-2024/tree/main/data/sera_metadata
- The H3N2 strains in the library listed in CSV format: https://github.com/jbloomlab/flu_seqneut_H3N2_2023-2024/blob/main/data/H3N2library_2023-2024_strain_order.csv
- The H3N2 strains in the library listed in FASTA format:
  - The trimmed HA1 sequences: https://github.com/jbloomlab/flu_seqneut_H3N2_2023-2024/blob/main/non-pipeline_analyses/library_design/results/2023-2024_H3_library_protein_HA1.fasta
  - The HA ectodomain sequences (with the H3 transmembrane domain removed): https://github.com/jbloomlab/flu_seqneut_H3N2_2023-2024/blob/main/non-pipeline_analyses/library_design/results/2023-2024_H3_library_protein_HA_ectodomain.fasta
  - The chimeric HA protein construct sequences with upstream signal peptide fixed to WSN sequence, downstream transmembrane domain fixed to H3 consensus, and downstream C-terminal tail fixed to WSN sequence: https://github.com/jbloomlab/flu_seqneut_H3N2_2023-2024/blob/main/non-pipeline_analyses/library_design/results/2023-2024_H3_library_protein_constructs.fasta

### Human sera

Human sera were from individuals of different ages from Seattle Children's Hospital and two adult vaccine cohorts, one based in Philadelphia, United States of America, and the other based in Australia

(*Table 1*). The Seattle Children's Hospital sera were obtained from routine blood draws from children (ages 1–14) receiving medical care in December 2023, which was approved by the Seattle Children's Hospital Institutional Review Board with a waiver of consent. Limited, non-identifying information was obtained from electronic health records. The sera from an adult vaccine cohort based in Philadelphia were taken from adults (ages 22–74) on the day of and 28 days post-vaccination with a 2023–2024 Northern Hemisphere egg-based vaccine (FluLaval quadrivalent influenza virus vaccine from GlaxoSmithKline) between October and December 2023. This study was approved by the Institutional Review Board of the University of Pennsylvania under protocol number 849398. The sera from Australia-based adult vaccine cohort were taken from adults (ages 26–54) on the day of and 17- to 21-day post-vaccination with a 2024 Southern Hemisphere cell-based vaccine (Flucelvax quadrivalent influenza vaccine from CSL Seqirus) between April and May 2024.

Before use in sequencing-based neutralization assays, all sera were treated with receptor-destroying enzyme and heat-inactivated using a protocol described in *Lee et al., 2019*. Briefly, one vial of lyophilized receptor-destroying enzyme II (Seikan) was resuspended in 20 mL PBS and passed through a 0.22 µM filter. Then, 100 µL of each sera was incubated with 300 µL of receptor-destroying enzyme (constituting a 1:4 dilution) at 37°C for 2.5 hr and then 55°C for 30 min. Sera were then used immediately or stored at –80°C until use.

## Cell lines

The following cell lines were used across experiments: 293T (ATCC, CRL-3216), MDCK-SIAT1 (HPA Cultures, 05071502), and MDCK-SIAT1-TMPRSS2 (from *Lee et al., 2018*). These cell lines were tested for *Mycoplasma* at the Fred Hutch Cell Bank core and confirmed to be *Mycoplasma* negative.

## Selection of H3N2 strains for a library for sequencing-based neutralization assays

To identify representative circulating strains, we used H3N2 Nextstrain (*Hadfield et al., 2018*; *Hedges et al., 2006*) builds available in November 2023. These trees are subset by Nextstrain-defined clade, subclade, and derived haplotype, where a derived haplotype is a more fine-grained level of genetic subdivision than subclade and is defined as a subset of strains belonging to the same subclade that each share additional amino acid mutation(s) and have achieved some threshold of child strains. We filtered all derived haplotypes by collection date, retaining only those haplotypes with a strain sequenced within a 12-month time period of library design. We also selected additional haplotypes that had not yet achieved the given threshold of child strains to be defined as derived haplotypes but were sequenced at high frequency within a 12-month window. For each of the derived haplotypes, we selected a naturally occurring HA sequence with the lowest divergence from the derived haplotype consensus sequence. This yielded 62 strains, representing the circulating diversity of H3N2 HA in 2023. Additionally, the past decade of egg- and cell-passaged vaccine strains was added, constituting an additional 16 strains. To visualize the 78 strains contained in the library, we built interactive trees on a background of the Nextstrain 2-year tree available in November 2023 (https://nextstrain.org/groups/jbloomlab/seqneut/h3n2-ha-2023-2024). We also drew static versions of this tree for paper figures, as described below. Note that we use the standard H3 ectodomain numbering for all these strains.

## Analysis of library composition

To obtain the count and fraction of HA1 and HA ectodomain sequences matching circulating sequences (depicted in *Figure 2c*, *Figure 2—figure supplement 1a*), all available H3 HA sequences between June 2022 and June 2024 were downloaded from the GISAID (*Shu and McCauley, 2017*) EpiFlu database. Each sequence was counted as a match if that given sequence was within 1 amino acid mutation of a library HA1 or HA ectodomain sequence. The fraction matching was then obtained by dividing the count of matching sequences by all total sequences. Strain counts (https://github.com/jbloomlab/flu_H3_2023_seqneut_vs_growth/tree/main/results/strain_counts) and code for these analyses can be found here (https://github.com/jbloomlab/flu_H3_2023_seqneut_vs_growth copy archived at *Bloom and Kikawa, 2026*).

All trees (in *Figure 2*, *Figure 3*, *Figure 5*, and *Figure 3—figure supplement 2*) were drawn using the *baltic* Python package (https://github.com/evogytis/baltic; *Dudas et al., 2025*) and custom code

([https://github.com/jbloomlab/flu_seqneut_H3N2_2023-2024/tree/main/non-pipeline_analyses/draw_trees](https://github.com/jbloomlab/flu_seqneut_H3N2_2023-2024/tree/main/non-pipeline_analyses/draw_trees)).

## Design of an H3 HA barcoded construct

Influenza has a single-stranded negative-sense segmented RNA genome. For proper packaging of the viral genomic segments, there must be intact 'packaging signals' both upstream and downstream of the viral genome, encompassing both coding and non-coding regions (*Gao and Palese, 2009*; *Hutchinson et al., 2010*; *Figure 1—figure supplement 1a*). To modify the influenza HA segment to incorporate a unique 16-nucleotide barcode without disrupting coding sequence and maintaining proper packaging, we inserted the barcode after the stop codon at the end of the HA coding sequence and duplicated the packaging signal at the 5' end of the negative sense HA vRNA, as previously described (*Welsh et al., 2024*; *Loes et al., 2024*; *Bacsik et al., 2023*; *Gao and Palese, 2009*; *Li et al., 2005*; *Harvey et al., 2011*; *Harvey et al., 2010*; *Wu et al., 2017*; *Figure 1—figure supplement 1b*). The HA coding region itself was chimeric, with the 3' signal peptide and 5' C-terminal tail using the protein sequence of laboratory-adapted A/WSN/1933 (H1N1) sequence, and the HA ectodomain and transmembrane domain derived from an H3 sequence. While the HA ectodomain region varied across strains in our library, the transmembrane domain was fixed to an H3 consensus sequence. Additionally, the last 105 nucleotides of the HA coding sequence were synonymously recoded to reduce homology between this region and the duplicated packaging signal (see *Figure 1—figure supplement 1b* legend for more details), thus improving barcode retention (*Loes et al., 2024*). This strategy of using A/WSN/1933 (H1N1) packaging signals is thought to improve incorporation of HAs, as all other non-HA genes were from A/WSN/1933 (H1N1) (*Harvey et al., 2011*; *Wu et al., 2017*). We used A/WSN/1933 (H1N1) non-HA genes because this lab-adapted strain produces viral stocks from reverse genetics at high titers, and because the extensive laboratory passaging reduces its biosafety risk.

## Cloning an H3 HA barcoded plasmid library

We applied a cloning strategy similar to that described in *Loes et al., 2024*, with modifications for the H3 HA construct described above. Briefly, each of the selected H3 HA ectodomains was ordered from Twist Biosciences encoding sequence homology with upstream A/WSN/1933 (H1N1) HA signal peptide and transmembrane coding region. A separate barcoded fragment was generated by PCR encoding sequence homology with the HA transmembrane and C-terminal tail coding region and an Illumina Read1 sequence that immediately follows the incorporated barcode. These fragments were assembled in a three-segment assembly reaction with a plasmid backbone using HiFi Assembly Mastermix (NEB). The plasmid backbone was the same derivative of a pHH21 plasmid (*Neumann et al., 1999*) described in *Welsh et al., 2024*, which contains an upstream A/WSN/1933 (H1N1) signal peptide, a downstream A/WSN/1933 (H1N1) partial endodomain and C-terminal domain, the 3' and 5' untranslated regions, the pHH21 promoter and terminator (*Welsh et al., 2024*), and the restriction enzyme sites XbaI and BsmBI (after the upstream and before the downstream packaging signals, respectively). The plasmid backbone was digested with restriction enzymes XbaI and BsmBI (NEB) per the manufacturer's instructions prior to the three-segment assembly. Each HA was cloned with 3 barcodes per strain and was verified by whole plasmid sequencing through Plasmidsaurus. The barcodes linked to each strain are listed here ([https://github.com/jbloomlab/flu_seqneut_H3N2_2023-2024/blob/main/data/viral_libraries/2023_H3N2_Kikawa.csv](https://github.com/jbloomlab/flu_seqneut_H3N2_2023-2024/blob/main/data/viral_libraries/2023_H3N2_Kikawa.csv)), and all HA variant plasmid maps in GenBank format are available here ([https://github.com/jbloomlab/flu_seqneut_H3N2_2023-2024/tree/main/non-pipeline_analyses/library_design/plasmids](https://github.com/jbloomlab/flu_seqneut_H3N2_2023-2024/tree/main/non-pipeline_analyses/library_design/plasmids)).

## Generating a barcoded influenza virus library

As in *Loes et al., 2024*, we used previously described reverse genetics approaches to generate viruses for all three barcoded HA plasmids for each influenza strain. Briefly, co-cultures of 5e5 293T cells and 5e4 MDCK-SIAT1-TMPRSS2 (*Lee et al., 2018*) cells were seeded in each well of a six-well plate in D10 media (DMEM supplemented with 10% heat-inactivated fetal bovine serum, 2 mM L-glutamine, 100 U per mL penicillin, and 100 μg per mL streptomycin). For each well, a master mix was made up of 250 ng of each non-HA segment (PB1, PB2, PA, NA, M, NP, NS) from A/WSN/1933 (H1N1) using WSN pHW18 plasmids (*Hoffmann et al., 2000*) and a given strain's HA plasmid pool (containing all

three of the independently barcoded strains). At ~24 hr after cell seeding, this plasmid mastermix was mixed with 100 µL of DMEM and 3 µL of BioT reagent, incubated for 15 min at room temperature, and then carefully added dropwise to wells. At 20 hr post-transfection, media was removed, cells were washed with PBS, and then 2 mL of influenza growth media (Opti-MEM supplemented with 0.1% heat-inactivated FBS, 0.3% bovine serum albumin, 100 µg per mL of calcium chloride, 100 U per mL penicillin, and 100 µg per mL streptomycin) was added. At 45 hr post-media swap (~65 hr post-transfection), supernatant was removed and centrifuged at 2000 rpm for 5 min to remove cell debris before being aliquoted for storage at –80°C. Before use, 100 µL of each of these virus stocks was passaged once more in 2 mL of influenza growth media, each in a single well of a six-well plate containing 4e5 MDCK-SIAT1-TMPRSS2 cells (*Lee et al., 2018*) for ~40 hr, as described in *Loes et al., 2024*. Supernatants were again cleared of cell debris by centrifugation at 2000 rpm for 5 min before being stored at –80°C.

## Assessing and correcting pool balancing

We again followed a similar approach to that outlined in *Loes et al., 2024*, to library pooling and balancing. Briefly, to generate a pool of HA strains with relatively equal strain balancing, we first created an equal volume pool of all barcoded HA strains. This equal-volume library was serially twofold diluted and used to infect MDCK-SIAT1 cells seeded at 5e4 cells per well in all wells of a 96-well plate. At ~16 hr post-infection, viral barcodes were sequenced as described below. Sequencing counts were then used to determine the relative contribution of each HA strain to the virus pool (*Figure 1— figure supplement 1d*). We then calculated the relative amount of each viral strain so that each strain's barcodes should correspond roughly to 1/78 of the sequencing counts for the total pool and re-pooled variants accordingly (*Figure 1—figure supplement 1d*). This re-pooled library was again serially diluted over MDCK-SIAT1 cells seeded at 1.5e5 cells per well in all wells of a 96-well plate, and viral barcodes were sequenced and used to verify that each strain (and each strain's constituent barcodes) was present at relatively equal amounts (*Figure 1—figure supplement 2e*).

## Determining infection conditions for sequencing-based neutralization assays

The virus library dilution and cell density used in our sequencing-based neutralization assays needed to maintain viral barcode counts in the linear range where viral transcription scales linearly with the amount of infectious virus added on cells, while maximizing the number of infectious virus particles added to each well (*Figure 1—figure supplement 2a and b*). As the H3 virus library was nearly double the size of the library described in *Loes et al., 2024*, we could not assume the same exact virus dilutions and cell densities would satisfy these requirements. Therefore, to identify the virus dilution and cell density conditions for the H3 library, we infected several cell densities (5e4 cells per well, 1e5 cells per well, and 1.5e5 cells per well) of MDCK-SIAT1 cells plated in influenza growth media (*Figure 1—figure supplement 2a*) with serially diluted library. At ~16 hr post-infection, we isolated and sequenced viral barcodes as described below. This approach allowed us to determine the transcriptional activity per µL of the library, rather than use a metric of viral infection (e.g. TCID50 per µL), which would be less relevant for an assay like ours that relies on vRNA for the main readout. We chose the 1:32 dilution of the virus library on the maximum density of cells tested, 1.5e5 MDCK-SIAT1 cells per well of a 96-well plate (*Figure 1—figure supplement 2b and d*) as it satisfied both of these outlined conditions.

## Sequencing-based neutralization assays

The experimental setup is nearly identical to that outlined in *Loes et al., 2024*. A few modifications to that general protocol were made for the increased library size and are noted accordingly. An updated protocol, including these modifications, is available here (https://dx.doi.org/10.17504/protocols.io. kqdg3xdmpg25/v2).

The initial steps of sequencing-based neutralization assays are similar to any 96-well plate neutralization assay. Human sera were prepared by treating with receptor-destroying enzyme and heat inactivation to prevent potentially confounding interactions with sialic acids contained in human serum as described above and previously (*Lee et al., 2019*). Using an initial dilution of 1:40 (accounting for the 1:4 dilution that was used during the receptor-destroying enzyme treatment), sera are then serially

twofold diluted across 11 columns of a 96-well plate in influenza growth media for a final volume of 50 µL diluted serum in each well. The final column of the 96-well plate was used for a no-serum control and contained only 50 µL of influenza growth media. A 50 µL volume of diluted virus library was then added to each well, and virus-serum mixtures were incubated at 37°C with 5% $CO_2$ for 1 hr. Following this incubation, 1.5e5 MDCK-SIAT1 cells were added per well in a total of 50 µL of influenza growth media and then incubated at the same conditions for ~16 hr. Here, and in the virus library titration experiments described above, MDCK-SIAT1 cells are used because they lack TMPRSS2 (which cleaves HA in producing cells, freeing and activating nascent virus particles for infection), and therefore should have only limited secondary viral replication (*Matrosovich et al., 2003*; *Böttcher et al., 2006*; *Böttcher-Friebertshäuser et al., 2010*). At ~16 hr post-infection, cells were lysed and barcodes were sequenced as described below.

## RNA extraction and barcode sequencing

Our sequencing protocol is similar to that described in Loes et al., except for one difference in the first round of PCR, which we note below. Briefly, supernatant is removed and cells are gently washed with PBS before each well is lysed with 50 µL of iScript RT-qPCR Sample Preparation Reagent (Bio-Rad) (*Shatzkes et al., 2014*) containing the barcoded RNA spike-in (which was generated, purified, and quantified as described in *Loes et al., 2024*) at 2 pM per well. The lysis reaction is allowed to proceed for 5 min before lysate is transferred off cells and into a new 96-well plate. We then use 1 µL of this lysate in 10 µL cDNA synthesis reactions for all wells using the iScript cDNA Synthesis kit (Bio-Rad) according to the manufacturer's instructions, using the same gene-specific primer as in *Loes et al., 2024*.

This cDNA product was then amplified in two rounds of PCR. In the first round, some sequencing preparations used the exact primers described previously (*Loes et al., 2024*). Other sequencing preparations used slightly modified forward primers which included a 6 bp index paired with the same reverse primer described in *Loes et al., 2024* (*5'-AGCAAAAGCAGGGGAAAATAAAAACAACC-3'*). The set of forward primers was as follows:

- 5'-GTGACTGGAGTTCAGACGTGTGCTCTTCCGATCTgctacaCCTACAATGTCGGATTTGTATTTAATAG-3'
- 5'-GTGACTGGAGTTCAGACGTGTGCTCTTCCGATCTatcgatCCTACAATGTCGGATTTGTATTTAATAG-3'
- 5'-GTGACTGGAGTTCAGACGTGTGCTCTTCCGATCTtgacgcCCTACAATGTCGGATTTGTATTTAATAG-3'
- *5'-GTGACTGGAGTTCAGACGTGTGCTCTTCCGATCTcagttgCCTACAATGTCGGATTTGTATTTAATAG-3'*

This allowed us to multiplex different plates using the same dual indices (added during the second round of PCR), which helped decrease sequencing costs. Regardless of which round one primer set was used, we used 5 µL of cDNA as template in a 50 µL reaction using KOD Polymerase Hot Start 2x Mastermix (Sigma) according to the manufacturer's instructions. The second round of PCR adding unique dual indexing primers was performed exactly as described in *Loes et al., 2024*, and after PCR samples were pooled at equal volume and ran on a 1% agarose gel at 85 V for 40 min. The bands were extracted and purified using Nucleospin Gel Extraction Kit (Takara) before being purified with Ampure XP beads (Beckman Coulter) and quantified by Qubit dsDNA High Sensitivity Kit (Thermo Scientific). The indexed, purified, and quantified DNA was then diluted to 4 nM and submitted for Illumina Sequencing, aiming for roughly 5e5–1e6 reads per well.

## Analysis of sequencing-based neutralization data

As in Loes et al., the analysis of sequencing-based neutralization data was performed using the modular *seqneut-pipeline* v3.1.3 (https://github.com/jbloomlab/seqneut-pipeline, *Bloom et al., 2024*), which calculates fraction infectivities from normalized barcode counts (as summarized in *Figure 1*), and then fits Hill curves to fraction infectivity values using the Python package *neutcurve* (https://github.com/jbloomlab/neutcurve, *Bloom and Davidsen, 2025*). The processing of these data is described fully in *Loes et al., 2024*, and in the *seqneut-pipeline* README (https://github.com/jbloomlab/seqneut-pipeline/README.md). All neutralization titers and custom visualization code are available at https://github.com/jbloomlab/flu_seqneut_H3N2_2023-2024 .

## Modeling strain growth

The analysis performing H3 HA strain growth rate estimates via multinomial logistic regression using the *evofr* (*Abousamra et al., 2024*) package is at https://github.com/jbloomlab/flu_H3_2023_seqneut_vs_growth (copy archived at *Bloom and Kikawa, 2026*). Briefly, we sought to make growth rate estimates for the strains in 2023 since this was the same timeframe when the sera were collected. To achieve this, we downloaded all publicly available H3N2 sequences from the GISAID (*Shu and McCauley, 2017*) EpiFlu database, filtering to only those sequences that closely matched a library HA1 sequence (within one HA1 amino acid mutation) and were collected between January 2023 and December 2023. If a sequence was within one HA1 amino acid mutation of multiple library HA1 proteins, then it was assigned to the closest one; if there were multiple equally close matches, then it was assigned fractionally to each match. We only made growth rate estimates for library strains with at least 80 sequencing counts (*Figure 5—figure supplement 1a*) and ignored counts for sequences that did not match a library strain (equivalent results were obtained if we instead fit a growth rate for these sequences as an 'other' category). We then fit multinomial logistic regression models using the *evofr* (*Abousamra et al., 2024*) package assuming a serial interval of 3.6 days (*Cowling et al., 2009*) to the strain counts. For the plot in *Figure 5a*, the frequencies are averaged over a 14-day sliding window for visual clarity, but the fits were to the raw sequencing counts. For most of the analyses in this paper, we used models based on requiring 80 sequencing counts to make an estimate for strain growth rates and counting a sequence as a match if it was within one amino acid mutation; see https://jbloomlab.github.io/flu_H3_2023_seqneut_vs_growth/ for comparable analyses using different reasonable sequence count cutoffs (e.g. 60, 50, 40, and 30, as depicted in *Figure 5—figure supplement 1*). Across sequence cutoffs, we found that the fraction of individuals with low neutralization titers and number of HA1 mutations correlated strongly with these multinomial logistic regression-estimated strain growth rates.

To determine which of the predictors (neutralization titers or HA1 mutations) most fully explains the dependent outcome variable (growth rate), we performed a multiple regression. However, the fraction of individuals with low neutralization titers and number of HA1 mutations were moderately collinear with a variance inflation factor of 2.7, and therefore the regression coefficients could not be interpreted as their relative contribution. To circumvent this, we assessed the importance of HA1 mutations versus neutralization titers by dropping each variable from the regression and then calculating the resulting difference in the regression mean squared error across 100 permutations of a leave-one-out multiple regression. Variables with a larger effect on mean squared error are assigned a greater importance (which is equal to the permutation mean squared error minus the base mean squared error). The relative importance of each variable was then found by summing each variable importance across 100 permutations and then dividing each variable's importance by the total importances from both variables. The notebook for the multiple regression and associated analyses can be explored here (https://github.com/jbloomlab/flu_H3_2023_seqneut_vs_growth/tree/main/non-pipeline_notebooks).

## Figures

Figures were generated with Vega-Altair v5.2 (https://altair-viz.github.io/) using custom Python code. Protein structures were visualized with UCSF ChimeraX v1.9 (*Meng et al., 2023*), and protein heatmaps were generated using the *byattr* command function. Figure panels were compiled and adjusted in Adobe Illustrator v27.0.

## Acknowledgements

We gratefully acknowledge all data contributors, i.e., the Authors and their Originating laboratories responsible for obtaining the specimens, and their Submitting laboratories for generating the genetic sequence and metadata and sharing via the GISAID Initiative, on which this research is based. An acknowledgement table for all sequences used in our analysis is available here (https://doi.org/10.55876/gis8.250304sc) and a table of GISAID accessions is available here (https://github.com/jbloomlab/flu_seqneut_H3N2_2023-2024/blob/main/non-pipeline_analyses/library_design/gisaid_acknowledgements/epi_for_episet.csv). This work was supported in part by the NIH/NIAID under award R01AI165821 to TB and JDB, contract 75N93021C00015 to SEH, JDB, and TB, and T32AI083203 to CK. JDB and TB are investigators of the Howard Hughes Medical Institute. Additionally, this research was supported by the Genomics & Bioinformatics Shared Resource, RRID:SCR_022606, of the Fred

Hutch/University of Washington/Seattle Children's Cancer Consortium (P30 CA015704), and by Fred Hutch Scientific Computing, NIH grants S10-OD-020069 and S10-OD-028685. Molecular graphics and analyses performed with UCSF ChimeraX, developed by the Resource for Biocomputing, Visualization, and Informatics at the University of California, San Francisco, with support from National Institutes of Health R01-GM129325 and the Office of Cyber Infrastructure and Computational Biology, National Institute of Allergy and Infectious Diseases.

## Additional information

### Competing interests

Andrea N Loes: ANL is an inventor on Fred Hutch licensed patents related to high-throughput techniques for characterizing viral antigenic variation. Scott E Hensley: SEH is a co-inventor on patents that describe the use of nucleoside-modified mRNA as a vaccine platform. SEH reports receiving consulting fees from Sanofi, Pfizer, Lumen, Novavax, and Merck. Jesse D Bloom: JDB is an inventor on Fred Hutch licensed patents related to high-throughput techniques for characterizing viral antigenic variation. JDB consults for Apriori Bio, GSK, Pfizer, and Invivyd on topics related to viral evolution. The other authors declare that no competing interests exist.

### Funding

| Funder | Grant reference number | Author |
| --- | --- | --- |
| National Institute of Allergy and Infectious Diseases | R01AI165821 | Trevor Bedford<br>Jesse D Bloom |
| National Institute of Allergy and Infectious Diseases | 75N93021C00015 | Scott E Hensley<br>Trevor Bedford<br>Jesse D Bloom |
| National Institute of Allergy and Infectious Diseases | T32AI083203 | Caroline Kikawa |
| Howard Hughes Medical Institute | | Trevor Bedford<br>Jesse D Bloom |

The funders had no role in study design, data collection and interpretation, or the decision to submit the work for publication.

### Author contributions

Caroline Kikawa, Conceptualization, Investigation, Methodology, Writing – original draft, Writing – review and editing; Andrea N Loes, Conceptualization, Investigation, Methodology, Writing – review and editing; John Huddleston, Software, Visualization, Writing – review and editing; Marlin D Figgins, Philippa Steinberg, Software, Writing – review and editing; Tachianna Griffiths, Elizabeth M Drapeau, Heidi Peck, Ian Barr, Janet A Englund, Resources, Writing – review and editing; Scott E Hensley, Resources, Funding acquisition, Writing – review and editing; Trevor Bedford, Software, Funding acquisition, Writing – review and editing; Jesse D Bloom, Conceptualization, Resources, Funding acquisition, Visualization, Methodology, Writing – original draft, Writing – review and editing

### Author ORCIDs

Caroline Kikawa ⓘ https://orcid.org/0000-0002-8654-5663
John Huddleston ⓘ https://orcid.org/0000-0002-4250-2063
Janet A Englund ⓘ https://orcid.org/0000-0003-1134-4178
Trevor Bedford ⓘ https://orcid.org/0000-0002-4039-5794
Jesse D Bloom ⓘ https://orcid.org/0000-0003-1267-3408

### Ethics

The Seattle Children's Hospital sera were obtained from routine blood draws from children (ages 1-14) receiving medical care in December 2023, which was approved by the Seattle Children's Hospital Institutional Review Board with a waiver of consent. Limited, non-identifying information was obtained from electronic health records. The sera from an adult vaccine cohort based in Philadelphia were

taken from adults (ages 22-74) on the day of and 28 days post vaccination with a 2023-2024 Northern Hemisphere egg-based vaccine (FluLaval quadrivalent influenza virus vaccine from GlaxoSmithKline) between October-December 2023. This study was approved by the Institutional Review Board of the University of Pennsylvania under protocol number 849398. The sera from Australia-based adult vaccine cohort were taken from adults (ages 26-54) on the day of and 17-21 days post vaccination with a 2024 Southern Hemisphere cell based vaccine (Flucelvax quadrivalent influenza vaccine from CSL Seqirus) between April-May 2024.

Reviewer #1 (Public review): https://doi.org/10.7554/eLife.106811.4.sa1
Reviewer #2 (Public review): https://doi.org/10.7554/eLife.106811.4.sa2
Author response https://doi.org/10.7554/eLife.106811.4.sa3

## Additional files

### Supplementary files
Supplementary file 1. Cell-passaged and egg-passaged H3N2 strains chosen for the seasonal human influenza vaccine each season, 2014–2024. For each Northern Hemisphere seasonal influenza vaccine from 2014 to 2024, we list the cell- and egg-based vaccine strains and the GenBank and GISAID identification numbers linked to each sequence. We also note the amino acid mutations present in each egg-passaged strain relative to its cell-passaged counterpart from the same vaccine season. Note the egg-passaged virus for the 2019–2020 season is listed (A/Kansas/14/2017X-327), but this strain did not grow to sufficiently high titers to be compatible with our assay and was excluded from the library.

MDAR checklist

### Data availability
The code for all the analyses performed here are freely available at https://github.com/jbloomlab/flu_seqneut_H3N2_2023-2024 (copy archived at *Kikawa and Bloom, 2026*), and there are links and descriptions of specific analyses and titer datasets available in the Methods section.

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
