## [Editor Report · eLife Assessment]

This **fundamental** study advances our understanding of population-level immune responses to influenza in both children and adults. The strength of the evidence supporting the conclusions is **compelling**, with high-throughput profiling assays and mathematical modeling. The work will be of interest to immunologists, virologists, vaccine developers, and those working on mathematical modeling of infectious diseases.

---

## [Referee Report · Reviewer #1 (Public review)]

The authors present exciting new experimental data on the antigenic recognition of 78 H3N2 strains (from the beginning of the 2023 Northern Hemisphere season) against a set of 150 serum samples. The authors compare protection profiles of individual sera and find that the antigenic effect of amino acid substitutions at specific sites depends on the immune class of the sera, differentiating between children and adults. Person-to-person heterogeneity in the measured titers is strong, specifically in the group of children's sera. The authors find that the fraction of sera with low titers correlates with the inferred growth rate using maximum likelihood regression (MLR), a correlation that does not hold for pooled sera. The authors then measure the protection profile of the sera against historical vaccine strains and find that it can be explained by birth cohort for children. Finally, the authors present data comparing pre- and post- vaccination protection profiles for 39 (USA) and 8 (Australia) adults. The data shows a cohort-specific vaccination effect as measured by the average titer increase, and also a virus-specific vaccination effect for the historical vaccine strains. The generated data is shared by the authors and they also note that these methods can be applied to inform the bi-annual vaccine composition meetings, which could be highly valuable.

Comments on revisions:

Thanks to the authors for the revised version of the manuscript. This version contains extended explanations clarifying the growth analysis by MLR. The other points of the initial report were addressed as well by language adjustments. As discussed during the revision process, future work might focus on the observed heterogeneity among the serum titers to different strains and its causes, which requires additional in-depth analysis.

---

## [Referee Report · Reviewer #2 (Public review)]

This is an excellent paper. The ability to measure the immune response to multiple viruses in parallel is a major advancement for the field, that will be relevant across pathogens (assuming the assay can be appropriately adapted). I only had a few comments, focused on maximising the information provided by the sera.

Comments on revisions:

These concerns were all addressed in the revised paper.

---

## [Author Response]

The following is the authors’ response to the previous reviews

**Public Reviews:**

**Reviewer #1 (Public review):**
The authors present exciting new experimental data on the antigenic recognition of 78 H3N2 strains (from the beginning of the 2023 Northern Hemisphere season) against a set of 150 serum samples. The authors compare protection profiles of individual sera and find that the antigenic effect of amino acid substitutions at specific sites depends on the immune class of the sera, differentiating between children and adults. Person-to-person heterogeneity in the measured titers is strong, specifically in the group of children's sera. The authors find that the fraction of sera with low titers correlates with the inferred growth rate using maximum likelihood regression (MLR), a correlation that does not hold for pooled sera. The authors then measure the protection profile of the sera against historical vaccine strains and find that it can be explained by birth cohort for children. Finally, the authors present data comparing pre- and post- vaccination protection profiles for 39 (USA) and 8 (Australia) adults. The data shows a cohort-specific vaccination effect as measured by the average titer increase, and also a virus-specific vaccination effect for the historical vaccine strains. The generated data is shared by the authors and they also note that these methods can be applied to inform the bi-annual vaccine composition meetings, which could be highly valuable.

We appreciate the reviewer’s clear summary of our work.

Thanks to the authors for the revised version of the manuscript. A few concerns remain after the revision:(1) We appreciate the additional computational analysis the authors have performed on normalizing the titers with the geometric mean titer for each individual, as shown in the new Supplemental Figure 6. We agree with the authors statement that, after averaging again within specific age groups, "there are no obvious age group-specific patterns." A discussion of this should be added to the revised manuscript, for example in the section "Pooled sera fail to capture the heterogeneity of individual sera," referring to the new Supplemental Figure 6.However, we also suggested that after this normalization, patterns might emerge that are not necessarily defined by birth cohort. This possibility remains unexplored and could provide an interesting addition to support potential effects of substitutions at sites 145 and 275/276 in individuals with specific titer profiles, which as stated above do not necessarily follow birth cohort patterns.

The reviewer is correct that there remains heterogeneity among the serum titers to different strains that we cannot easily explain via age group, and suggests that additional patterns could emerge. We certainly agree that explaining this heterogeneity remains an interesting goal, but as described in the manuscript we have analyzed the possible causes of the heterogeneity as exhaustively as possible given the available metadata. At this point, the most we can say is that the strain-specific neutralization titers are highly heterogeneous in a way that cannot be completely explained by birth cohort. We agree that further analysis of the cause is an area for future work, and have made all of our data available so that others can continue to explore additional hypotheses. It may be that these questions can only be answered by experiments on sera from newer cohorts where more detailed metadata on infection and vaccination history are available.

(2) Thank you for elaborating further on the method used to estimate growth rates in your reply to the reviewers. To clarify: the reason that we infer from Fig. 5a that A/Massachusetts has a higher fitness than A/Sydney is not because it reaches a higher maximum frequency, but because it seems to have a higher slope. The discrepancy between this plot and the MLR inferred fitness could be clarified by plotting the frequency trajectories on a log-scale.For the MLR, we understand that the initial frequency matters in assessing a variant's growth. However, when starting points of two clades differ in time (i.e., in different contexts of competing clades), this affects comparability, particularly between A/Massachusetts and A/Ontario, as well as for other strains. We still think that mentioning these time-dependent effects, which are not captured by the MLR analysis, would be appropriate. To support this, it could be helpful to include the MLR fits as an appendix figure, showing the different starting and/or time points used.

Multinomial logistic regression is a widely used technique to estimate viral growth rates from sequencing counts (PLoS Computational Biology, 20:e1012443; Nature, 597:703-708; Science, 376:1327-1332). As the reviewer points out, it does assume that the relative viral growth rates are constant over the time period analyzed. However, most of the patterns mentioned by the reviewer are not deviations from this assumption, but rather just due to the fact that frequencies are plotted on a linear scale. More specifically, our multinomial logistic regression implementation defines two parameters per variant: the initial frequency and the growth rate. The absolute variant growth rate is effectively the slope of the logit-transformed variant frequencies. Each variant's relative fitness depends on that variant's growth rate relative to a predefined baseline variant. Plotting frequencies on a logit scale does help emphasize the importance of the slope by showing exponential growth as a linear trajectory. We have added a new Supplemental Figure 9 that plots the frequencies from Figure 5A on a logit scale. As can be seen the frequency trajectories are closer to linear on the logit scale.

We have updated the results text to clarify the nature of the fixed relative growth rates per strain and to refer to this new supplemental figure as follows:

To estimate the evolutionary success of different human H3N2 influenza strains during 2023, we used multinomial logistic regression, which uses sequence counts to estimate fixed strain growth rates relative to a baseline strain for the entire analysis time period (in this case, 2023) [50–52]. Relative growth rates estimated by multinomial logistic regression represent relative fitnesses of strains over that time period. There were sufficient sequencing counts to reliably estimate growth rates in 2023 for 12 of the HAs for which we measured titers using our sequencing-based neutralization assay libraries (Figure 5a,b and Supplemental Figure 9). We estimated strain growth rates relative to the baseline strain of A/Massachusetts/18/2022. Note that these growth rates estimate how rapidly each strain grows relative to the baseline strain, rather than the absolute highest frequency reached by each strain. Each strain’s absolute growth rate corresponds to the slope of the strain’s logit-transformed frequencies at the end of the analysis time period (Supplemental Figure 9).

As the reviewer notes, the multinomial logistic regression implementation assumes a fixed growth rate for each strain over the time period being analyzed. This limitation causes the inferred growth rates to emphasize the latest trends in the analysis time period. For example, at the end of December 2023 in Figure 5A, the A/Ontario/RV00796/2023 strain is growing rapidly and replacing all other variants. Correspondingly, the multinomial logistic regression infers a high growth rate for that Ontario strain relative to the A/Massachusetts/18/2022 baseline strain. However, the A/Massachusetts/18/2022 strain was growing relative to other strains in the first half of 2023 since it has a higher growth rate than they do. However, there are modest deviations from linearity on the logit scale shown in the added supplementary figure likely because the assumption of a fixed set of relative growth rates over the analyzed time period is an approximation.

We have added the following text to the discussion to highlight this limitation of the multinomial logistic regression:

Our comparisons of the neutralization titers to the growth rates of different H3N2 strains was limited by the fact that only a modest number of strains had adequate sequence data to estimate their growth rates. Strains with more sequencing counts tend to be those with moderate-to-high fitness, which therefore limited the dynamic range of growth rates across strains we were able to analyze. Relatedly, the multinomial logistic regression infers a single fixed growth rate per strain for the entire analysis time period of 2023, and cannot represent changes in relative fitness of strains over that relatively short time period. Additionally, because the strains for which we estimated growth rates are phylogenetically related it is difficult to assess the statistical significance of the correlation [53], so it will be important for future work to reassess the correlations with new neutralization data against the dominant strains in future years.

(3) Regarding my previous suggestion to test an older vaccine strain than A/Texas/50/2012 to assess whether the observed peak in titer measurements is virus-specific: We understand that the authors want to focus the scope of this paper on the relative fitness of contemporary strains, and that this additional experimental effort would go beyond the main objectives outlined in this manuscript. However, the authors explicitly note that "Adults across age groups also have their highest titers to the oldest vaccine strain tested, consistent with the fact that these adults were first imprinted by exposure to an older strain." This statement gives the impression that imprinting effects increase titers for older strains, whereas this does not seem to be true from their results, but only true for A/Texas. It should be modified accordingly.

We agree with the reviewer’s suggestion that the specific language describing the potential trend of adults having the highest titers to the oldest strain tested could be further caveated. To this end, we have made the following edits to the portion of the main text that they highlighted:

Adults across age groups also have their highest titers to the oldest vaccine strain tested (Figure 6), consistent with the fact that these adults were likely first imprinted by exposure to an older strain more antigenically similar to A/Texas/50/2012 (the oldest strain tested here) than more recent strains. Note that a similar trend towards adult sera having higher titers to older vaccine strains was also observed in a more recent study we have performed using the same methodology described here [60].

Notably, this trend of adults across age groups having the highest titers to the oldest vaccine strains tested has held true in subsequent work we’ve performed with H1N1 viruses (Kikawa et al., 2025 Virus Evolution, DOI: https://doi.org/10.1093/ve/veaf086). In that more recent study, we again saw that adults (cohorts EPIHK, NIID, and UWMC) tended to have their highest titers to the oldest cell-passaged strain tested (A/California/07/2009), whereas children (cohort SCH) had more similar neutralization titers across strains. These additional data therefore support the idea that adults tend to have their highest titers to older vaccine strains, a finding that is also consistent with substantial prior work (eg, Science, 346:996-1000).

**Reviewer #2 (Public review):**
This is an excellent paper. The ability to measure the immune response to multiple viruses in parallel is a major advancement for the field, that will be relevant across pathogens (assuming the assay can be appropriately adapted). I only had a few comments, focused on maximising the information provided by the sera. These concerns were all addressed in the revised paper.

We thank this reviewer for the summary of our work and their helpful comments in the first revision.

**Reviewer #3 (Public review):**
The authors use high throughput neutralisation data to explore how different summary statistics for population immune responses relate to strain success, as measured by growth rate during the 2023 season. The question of how serological measurements relate to epidemic growth is an important one, and I thought the authors present a thoughtful analysis tackling this question, with some clear figures. In particular, they found that stratifying the population based on the magnitude of their antibody titres correlates more with strain growth than using measurements derived from pooled serum data. The updated manuscript has a stronger motivation, and there is substantial potential to build on this work in future research.Comments on revisions:I have no additional recommendations. There are several areas where the work could be further developed, which were not addressed in detail in the responses, but given this is a strong manuscript as it stands, it is fine that these aspects are for consideration only at this point.

We appreciate this reviewer’s summary of our work, and we are glad they feel the motivation is stronger in the revised manuscript.